# Active maintenance of proton motive force mediates starvation-induced bacterial antibiotic tolerance in *Escherichia coli*

Miaomiao Wang[1,2,3], Edward Wai Chi Chan[1,3], Yingkun Wan[2], Marcus Ho-yin Wong[1] & Sheng Chen [2✉]

Recent evidence suggests that metabolic shutdown alone does not fully explain how bacteria exhibit phenotypic antibiotic tolerance. In an attempt to investigate the range of starvation-induced physiological responses underlying tolerance development, we found that active maintenance of the transmembrane proton motive force (PMF) is essential for prolonged expression of antibiotic tolerance in bacteria. Eradication of tolerant sub-population could be achieved by disruption of PMF using the ionophore CCCP, or through suppression of PMF maintenance mechanisms by simultaneous inhibition of the phage shock protein (Psp) response and electron transport chain (ETC) complex activities. We consider disruption of bacterial PMF a feasible strategy for treatment of chronic and recurrent bacterial infections.

[1] State Key Laboratory of Chemical Biology and Drug Discovery, Department of Applied Biology and Chemical Technology, The Hong Kong Polytechnic University, Hung Hom, Kowloon, Hong Kong. [2] Department of Infectious Diseases and Public Health, Jockey Club College of Veterinary Medicine and Life Sciences, City University of Hong Kong, Kowloon, Hong Kong. [3] These authors contributed equally: Miaomiao Wang, Edward Wai Chi Chan. ✉email: shechen@cityu.edu.hk

Bacterial antibiotic tolerance is loosely defined as the ability to withstand the deleterious effects of antibiotics at concentrations that can otherwise be lethal, without exhibiting a change in antibiotic susceptibility upon re-growth under favorable conditions[1]. Recent studies reveal that the re-growth of antibiotic-tolerant cells, commonly known as persisters, that reside in the human body for a prolonged period is responsible for causing a wide range of chronic and recurrent infections, especially among immuno-compromised patients[2,3]. Hence, delineating the cellular mechanisms that underlie the onset and long-term maintenance of a stable antibiotic tolerance phenotype in bacteria is more clinically relevant than studying mechanisms governing the emergence of transient antibiotic tolerance in an exponentially growing population. Recent studies suggest the existence of redundant cellular mechanisms underlying tolerance formation and that dormancy alone is insufficient for long-term maintenance of the tolerance phenotype[4]. Nguyen et al.[5] showed that starvation-induced antibiotic tolerance involves curtailing the production of pro-oxidant metabolites and increasing anti-oxidant defenses. Intriguingly, it was found that persisters could be derived from the fastest-growing bacterial subpopulation, and that dormancy is neither necessary nor sufficient for the formation of persisters[4,6]. Efflux activities played an active role in conferring the bacterial tolerance phenotype through upregulation of the expression level of a range of multidrug efflux genes, such as *tolC, acrA, acrB, acrD, emrA, emrB, macA,* and *macB*[7]. In this previous study, time-lapse imaging of antibiotic-tolerant cells containing fluorescence-labeled TolC confirmed that high expression of *tolC* directly led to the formation of antibiotic-tolerant subpopulation. In view of these findings, we propose to investigate the full spectrum of regulatory and active defense mechanisms involved in the formation of bacterial antibiotic tolerance.

In order to unravel the range of active cellular mechanisms required for eliciting and maintaining prolonged phenotypic tolerance to antibiotics, nutrient starvation was chosen as an induction factor in this work because of the following reasons: (i) starvation was previously shown to induce a significantly stronger tolerance phenotype than compounds that inhibit bacterial growth under nutrient-rich conditions[8], suggesting that starvation-induced tolerance probably involves inducible defensive mechanisms and deserves more in-depth and systematic exploration; (ii) nutrient starvation is commonly encountered by bacteria during the infection process[9], hence starvation-induced tolerance responses should be investigated comprehensively at the transcriptional and physiological level; (iii) it is a readily manipulated test condition, which facilitates investigation into the key mechanisms underlying the formation of antibiotic tolerance[5,10,11].

In this work, we found that active maintenance of proton motive force (PMF) is essential for starvation-induced tolerance, and that disruption of PMF resulted in the eradication of the entire antibiotic-tolerant subpopulation. Verstraeten N. et al.[12,13] first reported that the Obg protein mediates tolerance formation through dissipation of PMF, and in a follow-up study, they reported that awakening persister cells and activation of their re-growth relied on PMF repolarization. In a separate study, it was reported that PMF is required for maintaining the survival of hypoxic non-growing *Mycobacterium. tuberculosis*[14]. High redox activity was found to yield a larger number of non-growing, tolerant cells; in addition, inhibition of respiration by deletion of genes that encode TCA cycle enzymes or by suppressing ETC activities that generate PMF was found to negatively affect tolerance formation[15]. The *caa₃*-encoded oxidase, which is highly efficient in creating a proton gradient, was found to enhance the survival fitness of *Pseudomonas aeruginosa* in nutrient starvation

conditions[16]. Likewise, bacteria became less tolerant to stress upon deletion of the *rgpF* gene, which resulted in PMF disruption[17]. Compounds that dissipate PMF were also reported to exhibit the ability to kill antibiotic-tolerant cells[18–21]. Taken together, these findings suggest that, although PMF dissipation may lead to tolerance formation, PMF may also be required for maintaining the viability of tolerant cells. The role of PMF on tolerant cells is complicated and needs further investigation. Our findings confirm that bacteria under starvation actively preserve PMF so as to maintain an antibiotic tolerance phenotype. Our findings provide the molecular basis for devising new approaches to eradicate antibiotic-tolerant subpopulation through PMF disruption.

## Results

**Genes in the *psp* family are overexpressed during starvation.** In order to explore the range of physiological responses that play an active role in the development and maintenance of antibiotic tolerance during starvation, we first performed RNA-Seq upon *Escherichia coli* BW25113 to identify genes whose expression level was upregulated even after the test organisms had experienced a prolonged starvation episode. As metabolic activities are reduced to a minimum when nutrients are depleted, the expression level of most functional genes is expected to be kept at a minimum, with the exception of essential proteins that may modulate adaptive physiological responses. Such proteins are therefore expected to contribute directly or indirectly to the formation of starvation-induced tolerance. Based on the RNA-Seq data, we identified the genes that, when compared with exponentially growing cells, were expressed at a level threefolds or more when the test organisms had encountered starvation stress for 24 h in physiological saline (Supplementary Data. 1). These genes included those encoding membrane transporters, transcriptional regulators, chaperone, DNA repair proteins and starvation stress sensors, among them, a functionally important gene cluster in which the expression level of all members was elevated was identified. This gene cluster is the *psp* family, which includes *pspA, C, E,* and *F* genes, the expression level of which was upregulated ~234-, 496-, 16-, and 3-folds, respectively, upon encountering starvation for 24 h (Supplementary Data 1). Products of the *psp* operon are known to be able to sense a change in PMF, membrane-stored curvature elastic stress, presence of mislocalized secretins and other deleterious factors; activation of the *psp* operon enables the bacterial cells to maintain PMF or avoid mislocalized secretin-induced toxicity[22,23]. We then focused on investigating the role of this gene cluster in mediating the expression of bacterial tolerance response during nutrient starvation.

**The role of Psp response in mediating starvation-induced tolerance.** To test whether products of the upregulated *psp* genes play a role in the formation of antibiotic tolerance, we monitored and compared the change in the level of starvation-induced tolerance of specific gene knockout mutants to the wild-type strain within a 6 days period and noticed that, although the level of tolerance in both the wild type and Δ*pspA* strain was similar at the initial phase of treatment, the proportion of ampicillin-tolerant cells in the Δ*pspA* mutant dropped at a significantly faster rate over the 6 days period when compared with the wild type. The size of the tolerant subpopulation in knockout mutants of the other genes in the *psp* family, however, was similar to that of wild type throughout the experiment (Fig. 1a; Supplementary Fig. 1a). The size of the antibiotic-tolerant subpopulation in the Δ*pspA* strain dropped to ~$3 \times 10^5$ cells/mL after 6 days of ampicillin treatment, which was only 10% of that of wild type

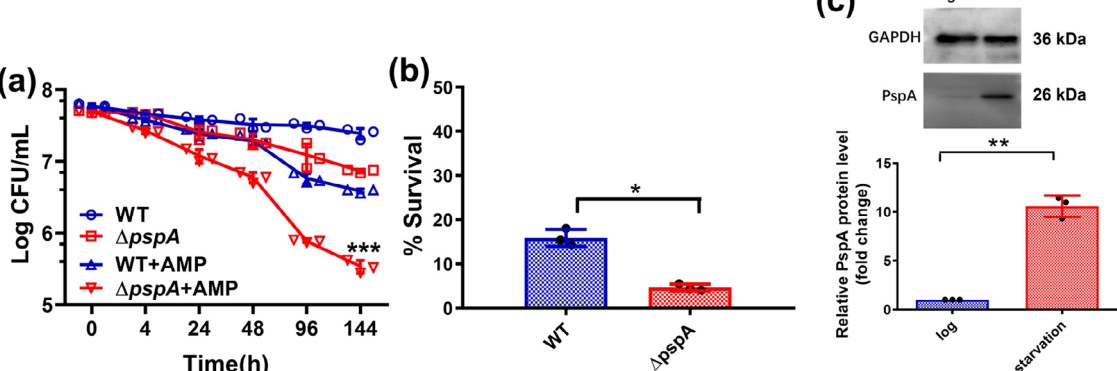

**Fig. 1 Activated Psp response during nutrient starvation affects bacterial survival and antibiotic tolerance. a** The wild-type *E. coli* BW25113 strain and the Δ*pspA* gene knockout strain were starved for 24 h, followed by treatment with ampicillin at 100 μg/mL for 144 h, variation in CFU recorded at different time points is shown. *P* value was tested between Δ*pspA* and Δ*pspA* + AMP at 144 h. **b** The survival rate of wild type or Δ*pspA* after 100 μg/mL ampicillin treatment 144 h compared with the non-treatment group. **c** Western blot analysis of the PspA protein in the bacterial population at a cell density of $OD_{600}$ 0.2, which had been subjected to starvation for 24 h, with the endogenous protein GAPDH as control. The relative expression level of the PspA protein recorded in the log phase population and those which had experienced nutrient starvation for 24 h upon normalization with the GAPDH control is shown. $n = 3$ biologically independent experiments. *indicates a *P* value of <0.05, **indicate a *P* value of <0.01, ***indicate a *P* value of <0.001 by two-tailed Student's test. Error bar represents standard deviation.

($\sim 4 \times 10^6$ cells/mL). The survival rate after ampicillin treatment for 144 h in the Δ*pspA* was ~5%, which was lower than that in wild type (~16%) (Fig. 1b). These findings imply that the *pspA* gene product is not essential for tolerance formation but is required for long-term maintenance of the tolerance phenotype. The size of the survival population after ampicillin treatment remained the same as that of the wild-type strain when a plasmid-borne *pspA* gene was introduced into the *pspA* gene deletion strain in gene complementation experiment (Supplementary Fig. 1b). We also confirmed that the phenotypes of the tolerant subpopulation were not due to genetic mutations that conferred drug resistance upon removal of antibiotic stress, as the tolerant subpopulation was able to regrow as antibiotic susceptible organisms, and the minimum inhibitory concentration (MIC) of knockout mutants of the *psp* gene family remained the same as that of wild type (8 μg/mL) (Supplementary Fig. 2; Supplementary Table. 1). PspA is the key functional protein among members of the Psp family and is known to play a major role in Psp response as it was involved in both preventing mislocalized secretin toxicity and maintaining the PMF, whereas the other proteins, such as PspB and PspC, were only involved in protection against mislocalized secretin toxicity[22,24]. We next investigated its role in the maintenance of the tolerance phenotype and tested whether an increase in gene expression of *pspA* actually resulted in a corresponding increase in protein level. Western blotting was performed, with results showing that PspA was barely detectable in the exponential phase, yet an abundance of this protein was synthesized upon encountering starvation for 24 h, the level of which was ~9.4 folds that of the exponential phase control (Fig. 1c; Supplementary Fig. 1c). In addition to maintenance of antibiotic tolerance, the *pspA* gene product was also found to have an effect on bacterial survival during starvation in the absence of antibiotics. Throughout a period of starvation for 6 days without ampicillin treatment, the population size of the *pspA* knockout strain shrank gradually to a level of $\sim 7 \times 10^6$ cells/mL, whereas that of wild type remained relatively constant at $\sim 2.5 \times 10^7$ cells/mL (Fig. 1a). On the other hand, we found that the size of the tolerant Δ*pspA* population during gentamicin treatment was only slightly less than that of the wild-type strain. The Δ*pspA* population which was tolerant to ciprofloxacin was similar to that of the wild-type strain, suggesting that the *pspA* gene product specifically enhances tolerance to β-lactams (Supplementary Fig. 1d).

**PspA protein plays a role in maintaining PMF during starvation**. It was reported that Psp proteins were involved in a wide range of membrane functions, with the PspBC complex being located in the inner membrane, interacting with PspA to prevent alteration in inner membrane permeability and cytoplasmic shrinkage[25]. We, therefore, hypothesized that deleting the *pspA* gene may undermine membrane integrity, leading to membrane leakage. By using the dye SYTOX Green to test membrane permeability during starvation, however, we showed that the amount of dye taken up by the wild type and Δ*pspA* strain during starvation was similar (Fig. 2a), indicating that membrane permeability was not significantly altered in the Δ*pspA* mutant. Likewise, although colistin treatment was found to cause membrane damage and an eventual increase in membrane permeability, the degree of changes in membrane permeability in both wild–type *E. coli* and Δ*pspA* mutant after colistin treatment were similar (Fig. 2a), suggesting that the Psp response conferred a little protective effect against this membrane destabilizing agent.

One major role of the PspA protein is to maintain bacterial PMF[26]. Oligomers of PspA other than the PspBCA complex were found to bind to membrane phospholipids and prevent proton leakage[23]. We then postulated that the reason why increased PspA expression could help maintain phenotypic tolerance is that it helped preserve PMF during starvation. We, therefore, used the dye $DiSC_3(5)$ to test the extent of changes in bacterial cell membrane potential upon entry into the starvation mode[27]. High-level accumulation of the dye in the bacteria cells would result in quenching of the overall fluorescence of the cell culture, whereas rapid release of the dye into the medium would result in dequenching upon depolarization of the dye[28]. In the exponential phase, the fluorescence intensities recorded for the wild type and Δ*pspA* strains were found to be similar with valinomycin as a positive control as it caused dissipation of membrane potential and then a sharp increase in fluorescence (Fig. 2b). Upon encountering starvation for 24 h, however, the fluorescence intensity of the Δ*pspA* strain was significantly higher than that of the wild type, indicating that the amount of dye accumulated intracellularly was much lower in the *pspA* mutant during starvation (Fig. 2c). On the other hand, the fluorescence intensity of the wild-type strain also remained at a similar level between the exponential phase and 24 h starvation, thereby confirming that

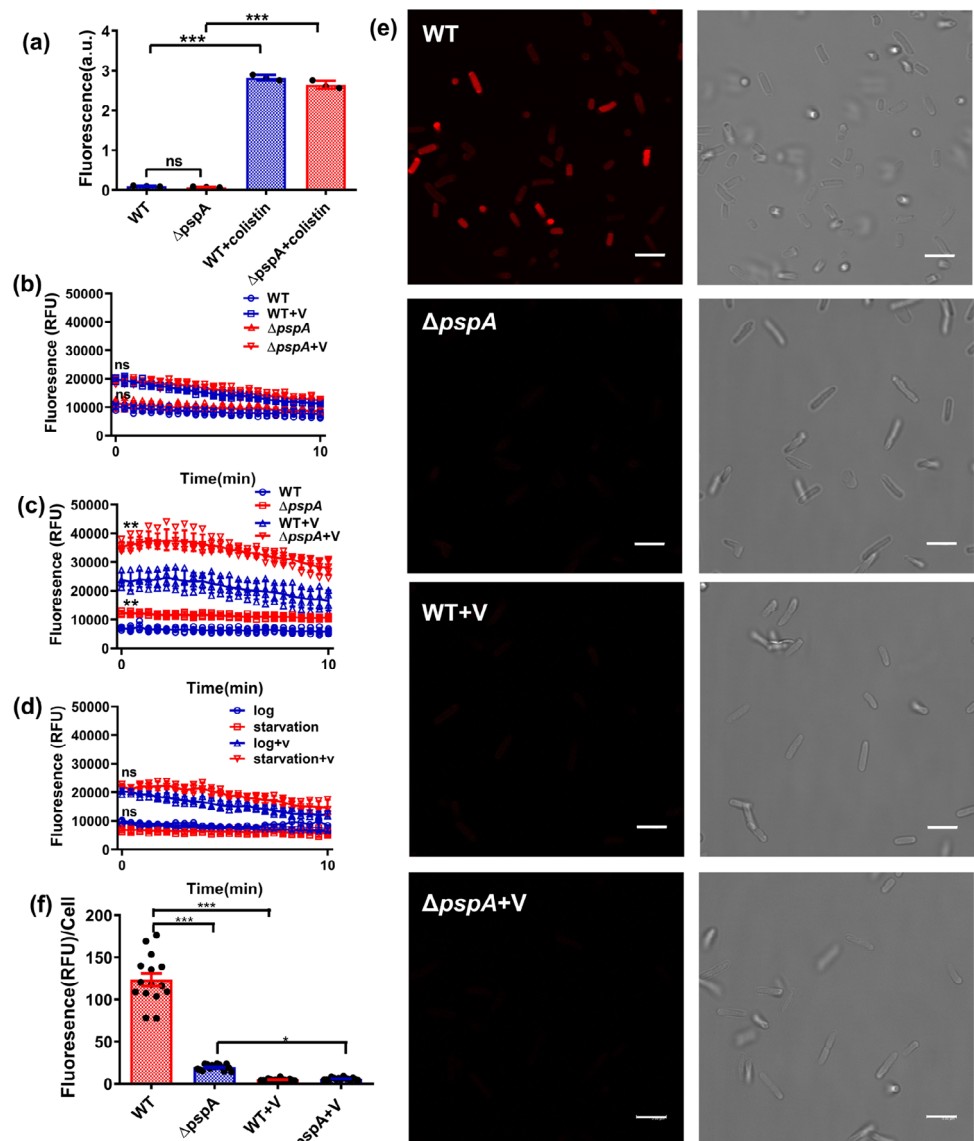

**Fig. 2 Psp response helps maintain PMF during starvation in *E. coli* without altering membrane permeability. a** Fluorescence intensity of SYTOX Green, which was used to detect membrane permeability of the wild type and Δ*pspA* strains, showed that the cell membrane in the Δ*pspA* strain remained intact; strains treated with colistin were included as a positive control. **b** Comparison between the fluorescence intensity of DiSC$_3$(5)-stained cells in the exponential growth phase of both wild type and Δ*pspA* strains reveals the same initial intensity and a similar degree of changes in membrane potential upon addition of valinomycin (labeled as V). *P* values were calculated between group (WT and Δ*pspA*; WT + V and Δ*pspA* + V) at the beginning time point. **c** Comparison between the fluorescence intensity of DiSC$_3$(5)-strained wild type and the Δ*pspA* strain, which had been subjected to starvation for 24 h depicts a much higher fluorescence intensity and hence a much lower PMF in the Δ*pspA* mutant. *P* values were calculated between group (WT and Δ*pspA*; WT + V and Δ*pspA* + V) at the beginning time point. **d** Comparison between the fluorescence intensity of DiSC$_3$(5)-stained exponentially growing wild type population and those, which had been subjected to 24 h starvation reveals a similar initial fluorescence intensity and also a similar degree of changes in fluorescence intensity, and hence membrane potential, upon addition of valinomycin. *n* > 3 biologically independent experiments. Error bar represents standard deviation. *P* values were calculated between group (WT and Δ*pspA*; WT + V and Δ*pspA* + V) at the beginning time point. **e** Confocal microscopy images of DiSC$_3$(5)-stained cells, which have been subjected to 24 h starvation in the absence and presence of valinomycin. The left and right panels are the fluorescence and bright-field images, respectively, (scale bar: 5 μm). **f** The mean DiSC$_3$(5) fluorescence intensity of the confocal microscopy image, which was calculated by the LAS X software. Data are the average of three observation field images. ns indicate no significance, * indicate a *P* value of <0.05, **indicate a *P* value of <0.01, ***indicate a *P* value of <0.001 by two-tailed Student's test. Error bar represents standard deviation.

PMF of the wild type could be maintained at a level equivalent to that of the exponential phase during starvation (Fig. 2d). These findings were consistent with the results of the confocal microscopy experiment, in which only the wild-type strain could be stained by DiSC$_3$(5) upon starvation for 24 h. The dye apparently could not enter cells of the Δ*pspA* strain because the membrane potential was too low (Fig. 2e). Based on the confocal images, we calculated the fluorescence intensity of each group and

found that the intensity of the wild-type strain (~100 RFU/cell) was approximately sevenfold that of the Δ*pspA* mutant strain (~15 RFU/cell); such finding further confirmed that knocking out the *pspA* gene would cause rapid dissipation of PMF during nutrient starvation (Fig. 2f). Our data indicated that PMF was maintained under starvation conditions as PspA was highly expressed and preserved PMF. This observation is consistent with the previous finding that PMF was advantageous for the survival

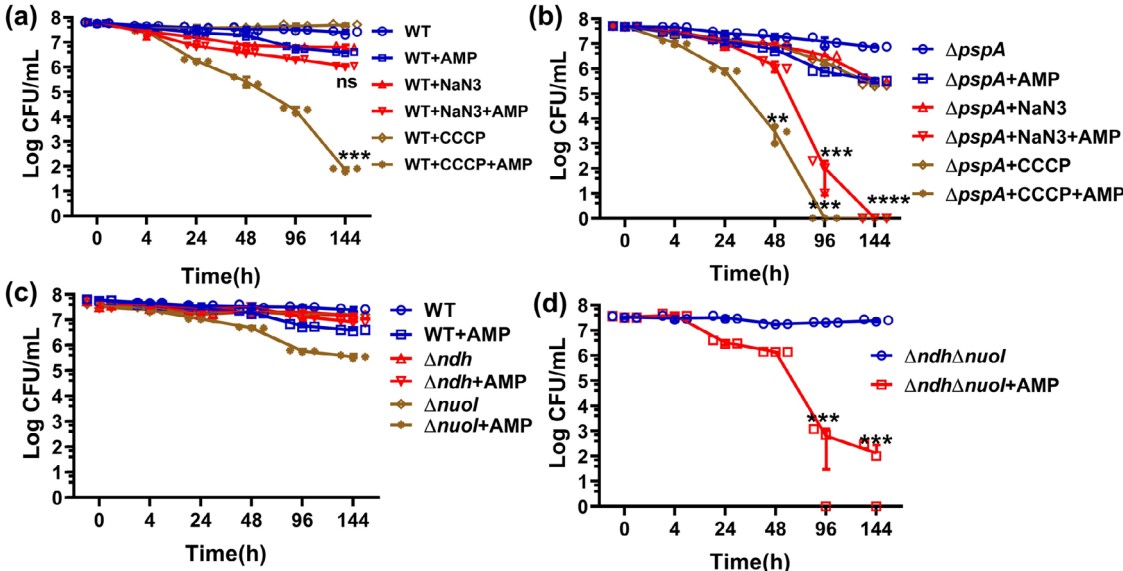

**Fig. 3 PMF is essential for maintaining the tolerance phenotype in bacteria under starvation. a** Population size of *E. coli* strain BW25113 was recorded at different time points upon starvation for 24 h, followed by treatment with ampicillin, sodium azide, CCCP, or various combinations of such compounds. *P* values were tested between WT + NaN3 and WT + NaN3 + AMP, WT + CCCP, and WT + CCCP + AMP at 144 h. **b** Population size of the Δ*pspA* gene knockout mutant recorded at different treatment time points upon starvation for 24 h and then treated with ampicillin, sodium azide, CCCP, or various combinations of such compounds. *P* values were tested between Δ*pspA* + NaN3 and Δ*pspA* + NaN3 + AMP, Δ*pspA* + CCCP and Δ*pspA* + CCCP + AMP at indicated time points. **c** *E. coli* strain BW25113 and the corresponding Δ*ndh* and Δ*nuoI* gene knockout mutants and double gene knockout mutant Δ*ndh*Δ*nuoI* **d** were starved for 24 h, followed by treatment with ampicillin for 144 h. Changes in population size during the course of 144 h are shown, along with data recorded in a no ampicillin treatment control. *n* = 3 biologically independent experiments. ns indicates no significance, \*\*indicate a *P* value of <0.01, \*\*\*indicate a *P* value of <0.001, \*\*\*\*indicate a *P* value of <0.0001 by two-tailed Student's test. Error bar represents standard deviation.

of hypoxic, non-growing bacteria or those under nutrient starvation conditions[14,16].

**Maintenance of PMF is essential for long-term survival of starvation-induced tolerant cells.** Upon identifying the PMF maintenance role of the *pspA* gene product and confirming the functional importance of PMF in actively maintaining phenotypic tolerance, we hypothesized that merely preventing dissipation of pre-existing PMF was not sufficient for totally abolishing the ability to maintain the tolerance phenotype as bacteria subjected to starvation stress still undergo a low level of oxidative phosphorylation to generate a basal level of PMF. To test this possibility, we determined whether sodium azide, which inhibits cytochrome C oxidase and hence the ability to generate PMF, could cause a reduction in the tolerance level of the bacterial population subjected to prolonged starvation[29]. Our results showed that the population size of the wild-type strain was only slightly reduced upon treatment with sodium azide, regardless of whether ampicillin was present or not (Fig. 3a). The effect of sodium azide alone on the Δ*pspA* mutant was similar to that of the wild-type strain, exhibiting a slight bactericidal effect. In the presence of ampicillin, however, sodium azide was able to eradicate the entire tolerant subpopulation at 144 h (Fig. 3b). These findings, therefore, confirm that, under nutrient-deficient conditions, PMF is essential for prolonged maintenance of the tolerance phenotype, as simultaneous inhibition of both PMF production and maintenance, by sodium azide treatment and deletion of the *pspA* gene, respectively, led to complete eradication of ampicillin-tolerant cells that formed under prolonged starvation conditions. To further confirm the role of PMF in starvation-induced tolerance response, the uncoupling agent carbonyl cyanide-m-chlorophenylhydrazone (CCCP), which is a known protonophore that destroys transmembrane proton

gradient, was used to test its effect on *E. coli* tolerance response[30]. When CCCP (1 μM) was added to *E. coli* cells that had been starved for 24 h, the size of the surviving population remained unchanged throughout the 144 h experiment, but the population size dropped to ~80cells/mL when ampicillin was present (Fig. 3a). When the concentration of CCCP was 10 μM, the size of the *E. coli* population recorded after CCCP treatment for 144 h dropped to the range of ~50 cells/mL and ~300 cells/mL regardless of whether ampicillin was present (Supplementary Fig. 3a). If the concentration of CCCP was increased to 100 μM, the size of the *E. coli* population which had been starved for 24 h dropped from ~2.5 × 10^7 cells/mL to ~5 × 10^5 cells/mL within 24 h, and to ~2 × 10^4 cells/mL if ampicillin was also present. The entire antibiotic-tolerant population was then eradicated by 48 h with or without ampicillin treatment, indicating that starvation-induced tolerant cells could no longer survive for a prolonged period upon the collapse of PMF (Supplementary Fig. 3a). Likewise, phenotypic tolerance to gentamicin cannot be maintained upon PMF dissipation (Supplementary Fig. 3b, c). However, tolerance to ciprofloxacin was not affected by CCCP treatment (Supplementary Fig. 3d, e). Taken together, our data showed that active maintenance of PMF is the key mechanism underlying prolonged expression of phenotypic antibiotic tolerance during nutrient starvation.

To confirm if active maintenance of PMF indeed has a key role in expressing phenotypic tolerance in bacteria, we further tested if disruption of the cellular mechanisms governing PMF formation could affect tolerance formation. ETC has an important role in generating PMF. Two enzymes, namely NADH dehydrogenase I and NADH dehydrogenase II, which are encoded by the genes *nuoI* and *ndh* respectively, are key components of the ETC[27,31]. Upon starvation for 24 h and then 6 days of ampicillin treatment, the population size of the

*E. coli* strains BW25113::$\Delta nuoI$ and $\Delta ndh$ was found to drop to ~$3.5 \times 10^5$ cell/mL and ~$8 \times 10^6$ cell/mL, respectively (Fig. 3c). Importantly, the size of the tolerant population of the *E. coli* BW25113::$\Delta ndh\Delta nuoI$ strain, in which both genes were simultaneously deleted, dropped sharply to ~200 cells/mL, a ~$10^4$ fold reduction, upon treatment with ampicillin for 6 days, suggesting that inhibition of activities of ETC components indeed severely affects production and maintenance of bacterial PMF, and hence the long-term survival of tolerant cells that are exposed to ampicillin (Fig. 3d; Supplementary Fig. 3f).

**Active efflux driven by PMF contributes partially to the maintenance of starvation-induced tolerant cells in *Escherichia coli*.** PMF is involved in numerous cellular functions; in particular, it has an essential role in maintaining efflux activities. Bacterial efflux could lead to a decrease in antibiotic accumulation, thereby facilitating the cells to form tolerant cells and survive from antibiotic treatment[7,32]. We, therefore, tested if the role of PMF in maintaining the antibiotic tolerance phenotype was owing to its effect on promoting efflux activities. We used a fluorescent β-lactam antibiotic known as BOCILLIN™ FL Penicillin (BOCILLIN) to depict the degree of accumulation of β-lactam antibiotics in the presence and absence of CCCP. We first confirmed that CCCP had little effect on the overall fluorescence signal as the fluorescence level exhibited by CCCP itself was only ~250 RFU, or ~180 times less than that of BOCILLIN (~45,000 RFU) (Supplementary Fig. 4a). In this experiment, we performed flow cytometry to assess the degree of accumulation of BOCILLIN with or without CCCP treatment. Wild-type bacterial cells that have been subjected to starvation for 24 h, followed by CCCP treatment, were generally well-stained by BOCILLIN with 97.86% of the cells exhibiting high BOCILLIN intensity (>$10^3$ RFU), whereas the fluorescence level recorded in the absence of CCCP was lower, with only 10.45% of the cells exhibiting high BOCILLIN intensity (>$10^3$ RFU), indicating that the amount of β-lactam antibiotic accumulated intracellularly increased upon PMF dissipation (Fig. 4a, b; Supplementary Fig. 4b–d). The percentage of $\Delta pspA$ cells that exhibited a high BOCILLIN level was 18.52%, which was more than that of the wild type (10.45%) (Fig. 4a, c; Supplementary Fig. 4c, e). The percentage of cells which exhibited high BOCILLIN intensity in the wild type and $\Delta pspA$ strain was similar after treatment with CCCP alone. However, the percentage of cells which exhibited extremely high BOCILLIN intensity (>$2 \times 10^4$ RFU) in the CCCP-treated $\Delta pspA$ strain (25.34%) was higher than that in the CCCP-treated wild-type sample (9.76%) (Fig. 4b, d; Supplementary Fig. 4d, f). This finding shows that the level of intracellular β-lactam antibiotic accumulation increases if PMF cannot be properly maintained and that the size of the tolerant subpopulation decreased when the intracellular BOCILLIN concentration increases. Consistently, the size of the tolerant population in $\Delta pspA$ was much smaller after treatment with both CCCP and ampicillin, when compared with treatment with ampicillin alone (Fig. 1a; Fig. 3a, b). We then determined whether the accumulation of BOCILLIN associated with artificial dissipation of PMF during starvation was due to failure to undergo efflux. The dye Nile Red, a common substrate of efflux pumps which only exhibits fluorescence in the intracellular compartment, was used in the investigation of bacterial efflux activities. In this experiment, Nile Red was incubated with the 24 h-starvation population for 30 mins, followed by the addition of CCCP and fluorescence measurement. The results showed that collapse or suppression of PMF upon addition of CCCP or deleting *pspA* gene, respectively, correlated well with an increase in fluorescence

signal, suggesting that the rate of extrusion of the dye out of the cells was reduced when efflux activities were suppressed. Specifically, the fluorescence intensity of the CCCP-treated wild-type strain increased to ~330,000 RFU at 30 mins, whereas the fluorescence intensity of the untreated strain was at the level of ~180,000 RFU, confirming that efflux efficiency in tolerant cells is reduced when PMF is abolished (Fig. 4e).

Tolerance formation was previously shown to negatively correlate with intracellular β-lactam accumulation[7]. To further determine whether efflux activities were indeed involved in starvation-induced antibiotic tolerance, we tested whether deleting the *tolC* gene, the product of which constitutes a key component of several major efflux systems, such as AcrAB-TolC and EmrAB-TolC, resulted in a reduction in the size of the antibiotic-tolerant population recorded during starvation. Under our assay condition, the size of the tolerant population in the *E. coli* $\Delta tolC$ mutant (~$5 \times 10^4$ cells/mL) was much smaller than that of wild type (~$2.5 \times 10^7$ cells/mL) upon treatment with ampicillin for six days, suggesting that efflux pumps played a role in the expression of the antibiotic tolerance phenotype (Fig. 4f; Supplementary Fig. 4g). This idea was further confirmed by testing the effect of PAβN, an efflux pump inhibitor, which was also found to cause a significant drop in the size of the tolerant population (~$1.5 \times 10^5$ cells/mL) (Fig. 4f). We showed that PAβN did not exert any negative effect on bacterial growth, indicating that the tolerance suppression effects conferred by this compound were not owing to its bactericidal effect (Supplementary Fig. 4h). These data suggested that efflux pump activity, which might be maintained through the PMF, contributed to the expression of phenotypic antibiotic tolerance during nutrient starvation.

Through comparison between the effect of PMF dissipation and efflux suppression on the survival of starvation-induced tolerant cells, we found that disruption of PMF exhibited a much stronger effect on tolerance suppression than inhibiting efflux activity. The entire tolerant cell population in the wild-type strain was recorded as ~50 cells/mL after 6 days of treatment with CCCP and ampicillin; in the case of *pspA* knockout, treatment with sodium azide/CCCP and ampicillin could cause complete eradication by 144 h. On the other hand, the size of the tolerant population in the wild-type strain remained at ~$5 \times 10^4$ cells/m L upon deletion of *tolC* or treatment with efflux pump inhibitor PAβN (Fig. 4f). These data strongly suggest that the maintenance of PMF is a key mechanism underlying the maintenance of starvation-induced bacterial antibiotic tolerance and that the functional role of PMF probably lies in the regulation of efflux and other important membrane transportation activities, which warrant further investigation.

**Maintenance of PMF is a key tolerance mechanism in both Gram-negative and positive bacteria.** To determine whether maintenance of PMF is an active cellular mechanism universally employed by various bacterial species to promote tolerance formation, we tested whether CCCP could eradicate starvation-induced tolerant cells of major bacterial pathogens. Our data confirmed that, a low concentration of CCCP was sufficient to suppress or even completely abolish phenotypic ampicillin tolerance in *K. pneumoniae*, *S. aureus*, *A. baumannii*, and *S. typhimurium*, and that a higher concentration of CCCP (100 μM) could eradicate tolerant bacteria cells even in the absence of ampicillin (Fig. 5b–e). The tolerant subpopulation was eradicated by ampicillin in the presence of 100 μM CCCP in *P. aeruginosa* (Fig. 5a). This finding confirmed that PMF maintenance is a universal mechanism underlying prolonged expression of phenotypic antibiotic tolerance in most bacterial species.

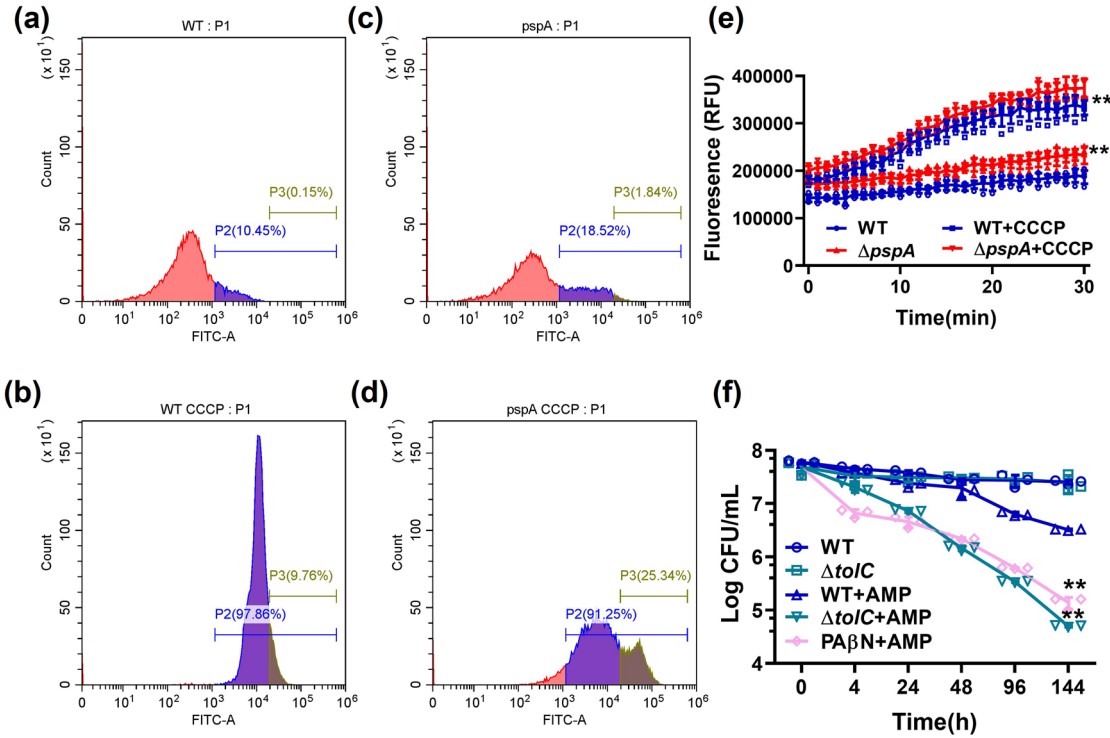

**Fig. 4 Active efflux driven by PMF contributes partially to the formation of an antibiotic-tolerant subpopulation during starvation. a–d** Fluorescence intensity recorded by flow cytometry depicts the degree of antibiotic accumulation (BOCILLIN^TM FL Penicillin, 10 μg/mL) in wild type and the Δ*pspA* strain when subjected to 24 h starvation in the presence and absence of CCCP. The P2 and P3 gates indicate the population whose BOCILLIN fluorescent intensity is >10$^3$RFU and 2 × 10$^4$RFU, respectively. **e** The fluorescent efflux substrate Nile Red was used to stain the wild type and Δ*pspA* bacterial population, which had been subjected to 24 h starvation in the presence and absence of CCCP. $n = 5$ biologically independent experiments. *P* values depicting the degree of significance in differences between the wild type and Δ*pspA*, and the wild type and wild type + CCCP samples at 30 mins. Error bar represents standard deviation. **f** Variation in the population size of *E. coli* strain BW25113 and the Δ*tolC* gene knockout strain, which had been subjected to starvation for 24 h, followed by treatment with ampicillin for 144 h. A no ampicillin treatment control of each of the BW25113 and Δ*tolC* gene knockout strains was included. The effect of the efflux pump inhibitor PAβN on starvation-induced ampicillin tolerance of the BW25113 strain is also depicted. $n = 3$ biologically independent experiments. *P* values were tested between WT and PAβN + AMP, as well as Δ*tolC* and Δ*tolC* + AMP at 144 h. **indicate a *P* value of <0.01, ***indicate a *P* value of <0.001 by two-tailed Student's test. Error bar represents standard deviation.

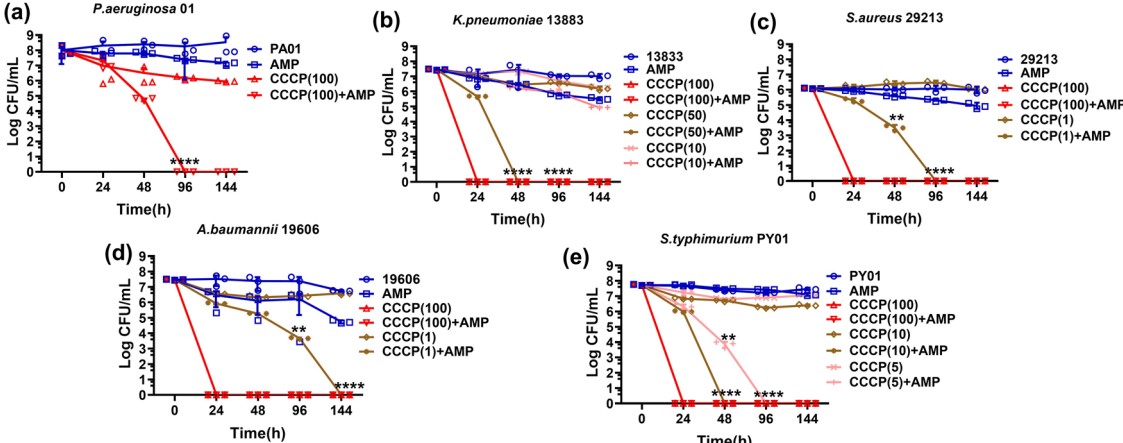

**Fig. 5 PMF maintenance is essential for starvation-induced tolerance formation in major Gram-negative and Gram-positive bacteria.** Changes in the size of antibiotic-tolerant subpopulation in *P. aeruginosa* (**a**), *K. pneumoniae* (**b**), *S. aureus* (**c**), *A. baumannii* (**d**) and *S. typhimurium* (**e**) which had been starved for 24 h, followed by treatment with 10× MIC ampicillin (AMP) alone (Supplementary Table. 2), CCCP alone and CCCP in the presence of 10× MIC ampicillin. CCCP(100), 100 μM CCCP; CCCP(50), 50 μM CCCP; CCCP(10), 10 μM CCCP; CCCP(5), 5 μM CCCP;CCCP(1), 1 μM CCCP. $n = 3$ biologically independent experiments. *P* values were tested between CCCP + AMP and CCCP with the same concentration at indicated time points. **indicate a *P* value of <0.01, ****indicate a *P* value of <0.0001 by two-tailed Student's test. Error bar represents standard deviation.

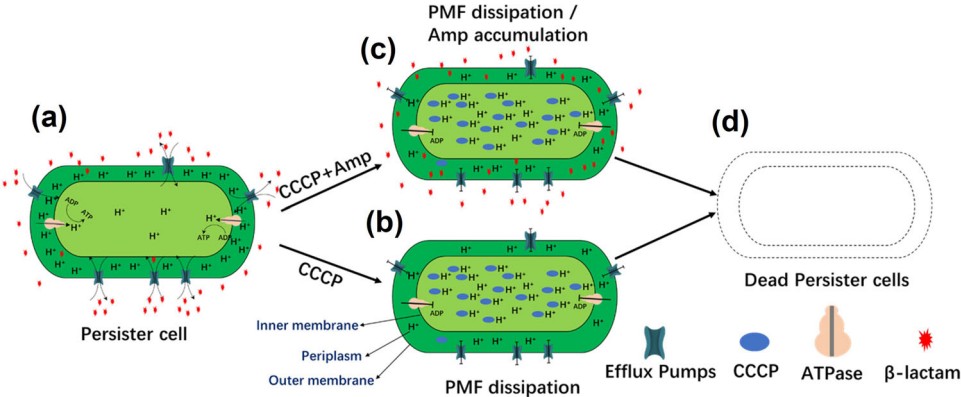

**Fig. 6 Proposed model of PMF-mediated development of starvation-induced tolerance. a** Maintaining PMF is essential for the prolonged survival of starvation-induced tolerant cells. Although bacteria may not be able to support a strong PMF when entering the tolerance phase, they nevertheless maintain a basic PMF level upon encountering starvation for a prolonged period. Efflux activities driven by PMF extrude β-lactams to facilitate tolerance formation; other membrane protein activities which presumably involve the import/export of specific metabolites/nutrients are supported by PMF and are also important for maintaining a tolerance phenotype. **b** Effect of PMF dissipators such as CCCP on tolerant cell killing. PMF dissipators cause rapid dissipation of bacterial membrane PMF and hence inhibition of ATP production, which in turn affects a series of cellular functions that are involved in maintaining the tolerance phenotype, leading to the killing of the tolerant cells. **c** Effect of PMF dissipator and ampicillin on tolerant cell killing. Tolerant cells are eradicated more effectively in the presence of β-lactam if PMF cannot be maintained under starvation stress. Dissipation of bacterial membrane PMF suppresses antibiotic efflux activities, leading to accumulation of antibiotics in the periplasm of tolerant cells. The accumulation of antibiotics and arrest of other cellular functions lead to the more effective killing of tolerant cells. **d** Complete dissipation of PMF with or without the presence of antibiotics results in the death of tolerant cells.

## Discussion

Antibiotic tolerance is the phenomenon in which a subpopulation of bacteria survive against lethal dosages of antibiotic treatment and regrow upon withdrawal of the drug. In this work, we aimed to delineate active tolerance mechanisms in bacteria. Through systematic analysis of the gene expression profile of bacteria subjected to prolonged starvation, we showed that products of the *psp* gene family played a role in preventing dissipation of PMF, thereby facilitating proper functioning of specific efflux and transportation systems even during nutrient starvation. We demonstrated that such cellular activities are essential for maintaining the survival fitness of the antibiotic-tolerant subpopulation. Discovered by Peter Model in 1990, the PspA protein was first shown to be induced in *E. coli* upon infection by the filamentous phage f1[33]. Psp proteins have since been postulated to play a role in regulating bacterial virulence, maintenance of PMF and mediation of envelope stress response[22]. The *rcsA* and *cpxP* genes, which mediate bacterial envelope stress response and were previously reported to play a role in maintaining PMF, were also found to be upregulated ~100 and 432-fold, respectively, in this study[34–36]. The Psp response was found to be involved in the regulation of indole-induced tolerance, as the indole-induced tolerance subpopulation size was reduced dramatically in the *pspBC* mutant[37]. It has also been shown that PspA was overexpressed in stationary phase bacterial population and that under alkaline conditions (pH 9), organisms lacking the *pspABC* genes exhibited a lower survival rate than the wild type, suggesting that the Psp response can enhance bacterial survival under hostile conditions[38]. Despite these findings, however, the functional importance of the Psp response in mediating the expression of phenotypic antibiotic tolerance in bacteria appears to be overlooked. This work describes the functional importance of PspA in mediating the expression of starvation-induced tolerance against β-lactams through maintaining PMF in bacteria.

In this work, the reason why we monitored changes in tolerance level over a 6 days period is that we believe the effect of lack of PMF maintenance function cannot be observed immediately. In fact, various previous studies showed that disrupting PMF and diminishing ATP levels could actually lead to the formation of tolerance, presumably by triggering dormancy[39,40]. There is currently no evidence that suggests that PMF is totally dissipated in tolerant cells; on the contrary, PMF is known to be required for the viability of non-replicating *M. tuberculosis*, as cell death was observed upon inhibition of activities of the ETC, which is essential for the generation of PMF[14]. It was also reported that tolerant cells were eradicated in the presence of compounds that cause dissipation of PMF[18,19]. Therefore, even though PMF dissipation is reported to lead to a reduced ATP level and trigger the onset of physiological dormancy in bacteria, PMF remains indispensable for the prolonged survival of dormant cells. The tolerance-mediating mechanisms are complicated as several lines of evidence show that dormancy is not sufficient or even essential for tolerant cell formation, as tolerant cells that formed in bacterial populations with high respiration activity or actively dividing cells were identified[4,6,15]. Orman et al. reported that the size of the tolerant subpopulation among bacteria with high respiration activity was actually higher than that in cells with low respiration activity, and that inhibition of ETC or the TCA cycle prevented tolerance formation[15]. Our work shows that, although dissipation of PMF could trigger tolerance formation even in the absence of starvation stress, a basal level of PMF is actually required for prolonged survival of bacterial tolerant cells. Hence, a lack of the ability to maintain PMF, as in the case of *pspA* knockout, results in a gradual reduction in the size of β-lactams tolerant subpopulation when compared with the wild-type strain. Inhibition of the ability to generate PMF by treatment with sodium azide also mildly affected tolerance to β-lactams. Importantly, when the ability to generate and maintain PMF was simultaneously inhibited, by treating the *pspA* knockout mutant with sodium azide, the β-lactams tolerance level was found to drop drastically. These observations, therefore, confirm that active maintenance of a basal level of PMF is required for the expression of phenotypic tolerance to β-lactams during nutrient starvation (Fig. 6a, c, d). Consistently, we found that knocking out key components of the respiratory ETC (Δ*nuoI*Δ*ndh*), which has a role in generating PMF, resulted in a dramatic drop in tolerance level. Our findings regarding the functional importance of PMF in the maintenance of tolerance to β-lactams are also consistent

with that of Ma et al.[41] who showed that inhibition of energy production by introducing mutations in the *sucB* and *ubiF* genes would affect the tolerance level. Taken together, it is highly likely that a basal level of metabolism is maintained in β-lactam-tolerant cells for ATP production and preservation of PMF, possibly through actively scavenging cellular materials released from dead cells as carbon sources. Our gene expression data showed that expression of various membrane-bound transporters was upregulated upon prolonged starvation (Supplementary Data 1).

Our works also showed that PMF maintenance was coupled to efflux activities that were also inducible to enhance bacterial survival fitness during starvation (Supplementary Data 1). These efflux activities are presumably involved in the export of intracellular antibiotics or toxic metabolites during starvation or other stresses, reducing the amount of antibiotic accumulated intracellularly and enabling organisms under starvation to become antibiotic tolerant[42]. We showed that efflux systems would lose the driven energy and exhibit decreased efflux efficiency if PMF collapsed. Consistently, Wu et al.[43] showed that structural defect of the AcrAB-TolC pump was associated with reduced antibiotic tolerance. Pu et al.[7] reported that efflux activities were involved in stationary phase-induced tolerance but the underlying regulatory mechanisms were not elucidated. On the other hand, this previous study also showed that specific efflux pumps conferred antibiotic tolerance, but whether such efflux activities are starvation-inducible was not shown. In this work, we confirmed that the efflux system played a role in maintaining phenotypic tolerance against β-lactams under prolonged starvation conditions. Our findings, therefore, help bridge this knowledge gap and have important implications in future exploration of starvation-induced tolerance mechanisms and the development of anti-tolerance strategies.

Our data confirm that the role of PMF is not limited to supporting efflux activity, as PMF dissipation as a result of treatment with CCCP alone leads to rapid eradication of tolerant cells, whereas deletion of efflux genes or treatment with efflux pump inhibitor only resulted in a moderate reduction in the size of tolerance population (Fig. 6a, b, d). PMF is essential for the proper functioning of a wide range of membrane proteins, including the aforementioned nutrient scavenging transportation proteins; PMF-dependent mechanisms underlying the maintenance of tolerance phenotype remain to be identified. Nevertheless, owing to its functional importance in maintaining the viability of tolerant cells, we consider PMF as an excellent target for the eradication of tolerant cells. Complete eradication of tolerant cells can rarely be achieved by inhibiting one specific cellular function. There were two previous reports of complete eradication of antibiotic-tolerant cells in Gram-positive bacteria, which involved the use of the retinoid and acyldepsipeptide antibiotic to inflict membrane damage and activate casein lytic proteases, respectively[44,45]. However, these antibiotics are not effective on Gram-negative organisms. Disrupting bacterial PMF to completely eradicate tolerant cells of both Gram-positive and negative bacteria possesses the clinical advantage.

To summarize, our study shows that PMF is essential for prolonged expression of starvation-induced tolerance phenotype against β-lactams in both Gram-positive and negative bacteria. Findings in this work represent an advancement in understanding the cellular basis of the phenomenon of bacterial tolerance against β-lactams: the emergence of β-lactams tolerant population is owing to the combined effects of metabolic shutdown and activation of a range of PMF-dependent defense mechanisms in response to variation in environmental conditions, with the latter being particularly important for long-term maintenance of the tolerance phenotype.

Disrupting bacterial PMF should be an effective approach to eradicate bacterial antibiotic-tolerant subpopulation.

## Methods

**Strains and culture**. All knockout strains were derived from *E. coli* BW25113 and single knockout strains were obtained from the Coli Genetics Stock Center (USA) (Supplementary Table 3). Double knockout strains were constructed by the lambda red recombination approach, in which the plasmid pKD46 was used for expression of Red recombinase, which comprises the terminator downstream of frp exo; pKD4 was used to express kanamycin resistance, pCP20 was used for the expression of Flp recombinase[46]. Plasmid pBAD18-Kan was used in gene complementation. Luria-Bertani (LB) broth was used for all cultures unless stated otherwise. All the strains were grown at 37 °C with shaking at 250 rpm.

**Tolerance assay**. Upon reaching the exponential phase, bacteria were washed and resuspended in saline (0.9% NaCl), incubated at 37 °C under constant shaking (250 rpm) for 24 h, followed by treatment with ampicillin at a concentration of ~10× MIC (Supplementary Table 1 and 2) for 144 h (6 days), supplementing fresh ampicillin every 48 h. Standard serial dilution and plating on LB agar were performed before and after ampicillin treatment for 4 h, 2 days, 4 days, and 6 days to determine the fraction of the test population that survived at different time points during the course of treatment[8].

**RNA sequencing and analysis**. Fresh *E. coli* K-12 BW25113 colonies were inoculated into LB medium and grown overnight at 37 °C under constant shaking (250 rpm). The overnight culture was diluted 100-fold in LB broth and cultivated for about 1 h until the $OD_{600}$ value reached 0.2 (exponential phase). Aliquots of this exponential phase culture were washed and resuspended in saline, cultured at 37 °C under constant shaking (250 rpm), followed by incubation with 100 μg/mL ampicillin at 37 °C for 24 h. Total RNA of bacteria collected from the exponential phase and starvation phase was extracted by the RNeasy Mini Kit (Qiagen, Germany); rRNA was removed by using the Illumina Ribo-Zero Plus rRNA Depletion Kit; samples were sent to Beijing Genomics Institute (Hong Kong) for transcriptome sequencing. Raw reads were first mapped to the reference genome with Hisat2. These mapped reads were provided as input to Cufflinks, which produced one file of assembled transcripts for each sample. The assembly files were merged with the reference transcriptome annotation into a unified annotation by Cuffmerge, which was quantified by Cuffdiff to generate a set of expression data. Cuffdiff found reads that mapped uniquely to one isoform and calculated isoform abundances, fold changes, and *q* values. The normalization strategy used was RPKM (Reads Per Kilobase Million) and only the genes whose RPKM was above 5 were chosen for analysis.

**Western blot analysis**. Upon starvation for 24 h, bacteria were harvested by centrifugation and solubilized in sample buffer for 10 mins at 100 °C. Total cellular proteins were separated by sodium dodecyl sulfate-polyacrylamide gel electrophoresis and electroblotted onto polyvinylidene fluoride membrane (BIO-RAD 0.2 μM) using a semi-dry electroblotting apparatus (BIO-RAD). Membranes containing fractionated samples were first probed with anti-PspA (polyclonal rabbit source) or anti-GAPDH (Abcam) antibodies and then washed with tris-buffered saline and Tween 20. Washed membranes were re-blocked and probed with anti-rabbit antibodies simultaneously. Target protein bands were detected by measurement of chemiluminescence exhibited by the horseradish peroxidase substrate (EMD Millipore); relative band intensities of Western blots were calculated by ImageJ v1.29[47].

**Membrane permeability assay**. The membrane permeability or integrity of the test organisms was measured using SYTOX Green (ThermoFisher), which can enter the cell through the damaged cell membrane and bind to nucleic acid, generating fluorescence signals. *E. coli* BW25113 and its Δ*pspA* derivative at a concentration of $OD_{600}$ of 0.2, which had been subjected to 24 h starvation, were collected by centrifugation (6000 × *g*, 2 mins), washed twice, and resuspended in saline. SYTOX Green was then added to give a final concentration of 1 μM, followed by incubation for 30 min in the dark at room temperature. The relative fluorescence signal in the wild type and Δ*pspA* strain was measured by a Cary Eclipse Fluorescence Spectrophotometer (Agilent), with an excitation wavelength of 488 ± 10 nm and an emission wavelength of 523 ± 10 nm[48,49].

**Assessment of the effect of PAβN on bacterial growth rate**. The overnight culture of the *Escherichia coli* BW25113 strain was diluted 1:100 in LB Broth, followed by the addition of 100 μM PAβN; a sample in which the only saline was added was included as a negative control. $OD_{600}$ value was tested at different time points.

**Membrane potential assay**. The transmembrane electrical potential was measured by using a membrane potential-sensitive probe, DiSC_3(5). The bacterial population in either the exponential phase ($OD_{600}$ of 0.2) or under 24 h starvation

(resuspended in saline) were collected by centrifugation ($6000 \times g$, 2 mins), washed twice, and resuspended in phosphate-buffered saline (PBS; pH 7.4), and then adjusted to $OD_{600}$ of 0.2. KCl and $DiSC_3(5)$ were added until the final concentration of 100 mM and 1 μM was, respectively, reached, followed by incubation at room temperature for 25 mins in the dark to allow the dye to penetrate through the outer membrane and produce a quenching effect. Valinomycin (1 μM) was then added to the positive control group to transport $K^+$ into the cytoplasm, which resulted in depolarization. The fluorescence reading was monitored by using a Clariostar Microplate Reader (BMG LABTECH) at an excitation wavelength of $622 \pm 10$ nm and an emission wavelength of $670 \pm 10$ nm for 10 mins. Upon depolarization, the dye was rapidly released into the medium, resulting in dequenching and facilitating detection fluorometrically[28]. Confocal imaging was also conducted for testing the difference between the membrane electrical potential of the wild-type strain and the ΔpspA mutant. The sample preparation method is the same as that prior to testing with the Microplate Reader except for the last step. In brief, cells were washed with PBS three times before confocal observation to remove extracellular $DiSC_3(5)$ dye. Bacteria were encased in 1.5%(wt/vol) low-melting-point agarose and then imaged by the Leica TCS SP8 MP Multiphoton Microscope with a ×60 oil-immersion objective[50]. $DiSC_3(5)$ was excited by 638 nm laser and fluorescence were detected by HyD detector at emission wavelength $675 \pm 25$ nm. The images were acquired and analyzed by the Leica Application Suite X (LAS X) software.

**Assessment of effect of proton ionophore and sodium azide on starvation-induced tolerance**. To determine whether keeping a level of PMF is essential for maintaining a tolerance phenotype in starvation-induced tolerant cells, the effect of the uncoupling agent CCCP or sodium azide (5 mM) was each individually tested. The test agents were added to bacteria that had been subjected to starvation for 24 h, followed by incubation at 37 °C and treatment with ~×10 MIC ampicillin for 144 h. Standard serial dilution and plating on LB agar were performed on samples collected every 24 h to assess changes in the size of the subpopulation that survived during the treatment process. For each sample, a control that did not receive ampicillin treatment was included in the experiment.

**Antibiotic accumulation assay**. The overnight bacteria culture was diluted 100-fold in LB broth and cultivated for about 1 hr until the $OD_{600}$ value reached 0.2 (exponential phase). Aliquots of this exponential phase culture were washed and resuspended in saline, cultured at 37 °C under constant shaking (250 rpm) for 24 h, followed by the addition of CCCP (1 μM). After 5 mins, BOCILLIN™ FL Penicillin (10 μg/mL) was added and incubated at 37 °C with shaking at 250 rpm for 1 h. Upon washing twice with PBS, the fluorescence signal was measured by flow cytometry CytoFLEX (Beckman). Microorganisms were identified by FSC (forward scatter) and SSC (side scatter) parameters. Fluorescence intensity was measured at 488 nm excitation, 525 nm emission.

**Assessment of efflux activity**. A 10-mL portion of the bacterial population that had been subjected to 24 h starvation was centrifuged at $6000 \times g$ for 5 mins at room temperature. The pellet was resuspended in PBS containing 1 mM $MgCl_2$(PPB) and adjusted to $OD_{600}$ 0.2. Nile Red that fluoresces only weakly in aqueous solutions but becomes strongly fluorescent in nonpolar environments was added to produce a final concentration of 5 μM followed by incubation at 37 °C for 30 mins, with 250 rpm shaking. CCCP was then added to produce a final concentration of 100 μM; fluorescence was measured for a period of 30 mins by a Clariostar Microplate Reader at an excitation wavelength of $544 \pm 10$ nm and an emission wavelength of $650 \pm 10$ nm[51,52].

**Determination of minimal inhibitory concentrations (MICs)**. The MIC of ampicillin against *A. baumannii* ATCC19606, *K. pneumoniae* ATCC13883, *P. aeruginosa* PAO1, *S. aureus* ATCC29213, *S. typhimurium* PY01, and *E. coli* K-12 BW25113 and its gene knockout derivatives (obtained from the Keio collection) was determined by incubating freshly grown cultures (Mueller Hinton Broth (BD Difco, America)) with various concentrations of ampicillin for 16 h, recording the minimal concentration that inhibited bacterial growth and resulted in a lack of turbidity. Results were based on the average of at least three independent experiments and interpreted according to CLSI guideline[53].

**Statistics and reproducibility**. Statistical methods used in this work are described in the figure legends. Statistical analysis was performed by using the GraphPad Prism software version 7.00 (Prism). The averages are shown, with error bars indicating the SD. Two-tailed Student's $t$ test was used in all cases when $n > 6$ unless stated otherwise in the figures; ns not significant; $*p < 0.05$; $**p < 0.01$; $***p < 0.001$; $****p < 0.0001$.

**Reporting summary**. Further information on research design is available in the Nature Research Reporting Summary linked to this article.

## Data availability

NCBI Gene Expression Omnibus accession number for RNA Sequence: PRJNA751277. pKD4 (#45605) and pBAD18-Kan (#217568) are available on Addgene. pKD46 (PVT6005) and pCP20 (PVT6006) are available in the Life science market. All other data are available from the corresponding author on reasonable request.

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

## Acknowledgements
We thank Professor Martin Buck of Imperial College London for generously providing the anti-PspA antibody. Provision of the Keio Collection strains by Hirotada Mori is deeply appreciated. This study was funded by the Research Impact Fund from the Research Grant Council of the Government of Hong Kong SAR (R5011-18F) and NSFC/RGC grant (N_PolyU521/18).

## Author contributions
Conceptualization, M.M.W., E.W.C.C., and S.C.; methodology, M.M.W. and E.W.C.C.; investigation, M.M.W., E.W.C.C., Y.K.W. and M.H.Y.W.; resources, M.M.W., E.W.C.C., and S.C.; review and editing, M.M.W., E.W.C.C. and S.C.; supervision, S.C.

## Competing interests
The authors declare no competing interests.
