## [Transparent Peer Review File · Communications Biology]

Reviewers' comments:

Reviewer #1 (Remarks to the Author):

Summary

Wang et al explore the contribution of the proton motive force to antibiotic tolerance. The authors first show that a strain of *E. coli* with a deletion in *pspA* has a slight fitness disadvantage during survival in stationary phase and is more sensitive to ampicillin exposure over long durations in starvation conditions. PspA expression appears to be greater in stationary phase relative to active growth. Exponentially growing wild type *E. coli* and the *pspA* deletion had similar DiSC3(5) staining +/- valinomycin. Stationary phase *pspA* deletion *E. coli* had higher DiSC3(5) fluorescence +/- valinomycin relative to wild type. This suggests that the *pspA* strain had a diminished delta psi component of the PMF, since DiSC3(5) is uptaken in response to delta psi, thus quenching its own fluorescence. Next, the authors tested various chemical and genetic perturbants of the PMF and assessed tolerance to either ampicillin or gentamicin. An interesting observation was that deleting both *ndh* and *nuoI* resulted in significantly enhanced sensitivity to ampicillin in stationary phase, relative to wild type. Next, the authors explore the relevance of drug efflux (*tolC*) in antibiotic tolerance, showing that deletion of *tolC* enhances stationary phase sensitivity to ampicillin. Lastly, a small collection of bacterial species are assayed to determine whether PMF disruption potentiated ampicillin in stationary phase across different pathogens. Overall, the study could be of interest in building on a growing body of literature exploring the physiology of cells in antibiotic tolerant states. However, there are a number of issues that need to be resolved.

Comments

1. Most experiments shown in the paper use ampicillin as the bactericidal antibiotic. However, it would be interesting to test other conventionally bactericidal compounds to understand whether enhanced sensitivity is observed across discrete drug classes. For example, is *pspA* sensitive to killing by ciprofloxacin in stationary phase (could be added to figure 1). Is this strain more sensitive to gentamicin or colistin? Expanding the antibiotic collection used in studies of how gene deletion mutants and PMF-dissipating chemicals impact tolerance would improve the ability to generalize some of your comments, which are limited to ampicillin currently. Furthermore, ampicillin disrupts PG biosynthesis, which one could imagine would be more synergistic with PMF disruption due to both components being at the cell envelope.
2. The Western blot in figure 1b needs to be re-conducted and re-quantified since the loading control GAPDH is clearly in higher abundance in starvation than in log phase.
3. In figure 2, the authors use valinomycin as a PMF-disrupting molecule, showing that the *pspA* deletion strain displays higher DiSC3(5) fluorescence upon exposure to valinomycin. This molecule is a specific disruptor of the delta psi component of the PMF. How does the *pspA* strain respond to treatment with nigericin for example, which is a specific disruptor of the delta pH component of the PMF? Is the *pspA* strain equally sensitive to disruption of both components of the PMF, or just the delta psi?
4. Does valinomycin or nigericin potentiate antibiotics in deep stationary phase, like what you observe with CCCP or (to some extent) NaN₃?
5. Comparing figure 2a and figure 2b, it appears as if in this case the *pspA* knockout on its own (figure 2b) does not have a large fitness disadvantage relative to the wild type strain on its own (figure 2a). However, there is a clear fitness cost that we can observe in figure 1. Can the authors explain this discrepancy?
6. In figure 5, there is no evidence that combining CCCP with ampicillin is synergistic. In all cases except *P. aeruginosa*, CCCP is completely bactericidal on its own, and the CCCP + ampicillin curve simply overlays with the CCCP curve. This is unconvincing to show that PMF maintenance is important for antibiotic tolerance across bacterial species. This experiment needs to be optimized to clearly show that chemically or genetically modulating PMF can enhance bactericidal antibiotic efficacy against tolerant cells across different bacterial species.

7. 100 ug/ml ampicillin is a fairly high concentration to be working with. Is ampicillin activity not observed below this concentration in any of the genetic or chemical conditions? I suggest providing some dose-response data showing at which concentration (at some endpoint, 144 hrs perhaps) ampicillin potentiation is observed in the various genetic and chemical conditions.

8. Does a *pspA* *tolC* double deletion display enhanced sensitivity to ampicillin relative to either single-gene deletion strain, or does it simply look like the *pspA* strain, since presumably this would be the gene that would have more impact on *tolC* rather than vice versa.

Reviewer #2 (Remarks to the Author):

In this manuscript ("Active Maintenance of Proton Motive Force is Key Mechanism of Starvation-induced Bacterial Antibiotic Tolerance"), Wang and colleagues seek to demonstrate that maintenance of PMF is essential for prolonged antibiotic tolerance. The authors show that in *Escherichia coli* incubated in saline (0.9 or 0.85% NaCl) for prolonged periods of time, inhibition of the electron transport chain in these starved cultures substantially reduced survivor in the bacterial population, including cells that are tolerant to ampicillin. While the authors are addressing an important clinical concern—antibiotic-tolerant bacteria that underlie recurrent infections— a number of key controls are absent from their experiments and some of their experimental designs are flawed.

Experimental Comments:

- The authors should quantify ATP in their cells (e.g., using Promega's BacTitre Glo kit) to ensure that while PMF in some of their samples are dissipated (valinomycin, CCCP, sodium azide-treated samples), the cells are retaining enough adenylate charge to remain viable (Chapman et al., 1971, PMID: 4333317).
- In their RNA-Seq and antibiotic tolerance experiments, the authors inoculate *E. coli* cultures that had reached OD600 of 0.2 in saline (0.85% or 0.9% NaCl). As phage shock proteins (e.g., *pspA*) can be induced by prolonged stationary phase at high pH and osmotic shock, could the choice of "starvation" media contribute to the increased expression of *pspABCD*, as detected in their RNA-seq experiments? Why did the authors choose to use NaCl solutions as opposed to phosphate-buffered saline or defined media lacking a carbon source (e.g., M9 salts without glucose) as their starvation condition? It is unclear whether the starvation transcriptome that the authors report here are generalizable to the transcriptomes of cell starved under buffered conditions.
- The authors primarily focused on the overexpression of *psp* genes in their RNA-seq data, yet they do not comment on other up-regulated genes identified in their analysis. For instance, a few transporters and efflux pumps are as highly expressed as *pspABC*.
- The authors did not report on the genes that were down-regulated in their RNA-seq experiments.
- Fig. 1A: The authors did not report their CFU counts for $t=0$.
- The reduced culturability of the Δ *pspA* mutants by ~4-fold after 144 h of starvation could have contributed to the reduction in ampicillin persists throughout the course of treatment. Are the differences in survival between WT and Δ *pspA* cells without treatment statistically different? How about after treatment? As Δ *pspA* did not eradicate the population during starvation with and without ampicillin treatment, the title of Fig. 1 indicating that the Psp response is "essential" for survival and tolerance, is misleading.
- The authors should complement their deletion mutants.
- Fig. 1B: The authors should have tried to load similar amounts of proteins in each lane of the gel (the GAPDH band looks to be a few folds fainter in the Log sample relative to the Starvation sample), and they should show the full blot in the supplemental figures. The dark background observed in the GAPDH bands can potentially affect quantification using ImageJ.
- Fig. 2C: Perhaps the authors should add DiSC3(5) at the time of incubating the cells in saline and quantify the change in fluorescence in the population at different times thereafter. The way the experiment is set up now, it is unknown whether the fluorescence is higher in the Δ *pspA* mutants because of decreased dye uptake or increased dye release and dequenching.
- How were the relative fluorescence values determined? Why are the values lower for the log phase samples in Fig. 2D lower than those in Fig. 2B and the starvation samples lower in Fig. 2D compared with those in Fig. 2C.
- The microscopy images in Fig. 2E and 2F do not agree with the data from Fig. 2C-D. According to

Fig. 2C, fluorescence of the DiSC3(5) dye is quenched in WT cells, yet these cells were the brightest in the microscopy experiments. On the other hand, the valinomycin-treated samples were the most fluorescent when they were quantified using a plate reader, yet these populations were dark in the microscopy images. If, as the authors state in line 209, the "membrane potential is too low" in the Δ pspA mutants, how did the dye enter these mutants in Fig. 2C?

- Lines 244-246: Perhaps it is not surprising that starving bacteria maintain basal PMF, otherwise they would be deenergized and dead.
- Line 252: I disagree that the Δ pspA mutants "lacks the ability to maintain PMF" as the authors claim, as many of these cells remained viable after 144 h in saline.
- Fig. 3E and F: The authors should include WT cells treated with antibiotics (AMP/ GEN) alone as controls for comparison.
- Line 300: I also disagree with the authors' claims that their findings challenge the existing view of the impact of PMF suppression on dormancy and tolerance or previous findings that addition of glycolytic intermediates can enhance aminoglycoside-mediated killing. As the authors stated, they are treating *E. coli* with CCCP for a much longer duration than in previous studies. Furthermore, the authors never quantified ATP/ adenylate charge to ensure that their cells retained sufficient ATP/ adenylate charge to remain viable.
- Fig. 4: As the authors are looking at BOCILLIN uptake in a phenotypically heterogeneous population, it would have been better for them to quantify fluorescent cells by flow cytometry rather than using a plate reader.
- Fig. 4 A&B: If the fluorescence of the "No CCCP" sample is 2-fold lower than the "CCCP" sample, why is the fluorescence of the "no CCCP" control not detected at all in the microscopy images?
- Fig. 4 C&D: Is the Nile Red accumulation data and the survival of the mutants statistically different from the controls (no CCCP and WT controls)?
- Fig. 5: What is the MIC of ampicillin against *Salmonella*? Why would the authors test *P. aeruginosa*, *K. pneumoniae*, and *A. baumannii* when these species are naturally resistant to AMP? The data in panels B-F mainly shows that CCCP can lead to death of these organisms.

Additional Comments:

- Overall, the correct terminology should be "dissipation of PMF" and "depolarization of the membrane". "PMF depolarization", as the authors stated in line 85, is incorrect.
- Lines 84-87: In the description of the work by Verstraeten and colleagues, "yet in a follow-up study" should be replaced with "and in a follow-up study", because if membrane depolarization leads to dormancy, it makes sense that repolarization contributes to persisting awakening.
- On that note, Ref. 19 is incorrect, as the authors should be citing the works by Verstraeten and colleagues.
- Lines 95-96: It should be noted that the compounds tested in Refs. 17 and 18 do not only dissipate PMF, they can permeabilize the membrane (glycopolymers) or inhibit ATP synthase (diarylquinolines). As such PMF dissipation is not their only mechanism of killing.
- Why did the authors choose to use 0.9% NaCl for their tolerance assay, but used 0.85% NaCl for their RNA-seq experiments?
- As all of the PMF disruption methods that the authors used also disrupts mitochondrial PMF, their claim at the end of their abstract that disrupting bacterial PMF is a potential strategy for treating chronic infection may not actually be feasible.
- Additional details on their RNA-seq sample preparation is warranted. What was the depth of read? Was rRNA depletion carried out during library preparation?

There are a number of typographical errors throughout the manuscript. Here is a partial list:

Title: ("Active Maintenance of Proton Motive Force is Key Mechanism of Starvation-induced Bacterial Antibiotic Tolerance" should be changed to "Active Maintenance of Proton Motive Force is a Key Mechanism of Starvation-induced Bacterial Antibiotic Tolerance".

Main Text:

- Line 93: Replace "Pseudomonas Aeruginosa" with "Pseudomonas aeruginosa".
- Fig. 2B-D: The symbols are really difficult to discern. The authors should consider using larger symbols and using filled in vs. hollow symbols to compare their +/- valinomycin symbols.
- Line 281: Do the authors mean to say NADH dehydrogenase II? The symbol is replaced with a square in the document.

- Line 291: I am not sure that the authors are correct in saying that $\Delta ndh \Delta nuoI$ resulted in the "strongest tolerance suppression effect recorded". Deletion of some of the SOS response genes (e.g., *recA* and *recB*) results in a similar magnitude of decrease in persistence toward fluoroquinolones, even in non-growing populations.

Supplemental File:

- Line 29: "Pkd46" should be changed to "pKD46".
- Line 33: "rmp" should be changed to "rpm".
- On line 36, 44 and other locations: "oC" is not visible and a square is present instead.
- Line 45: "OD600" should be modified to "OD600".
- Line 79: Please remove the comma at the end of the sentence.
- Line 206: "DiSC3(5)" should be changed to "DiSC3(5)".

Reviewer #3 (Remarks to the Author):

Wang et al investigated the roles of PMF in antibiotics tolerance. The authors combined molecular microbiology techniques with fluorescence measurements of membrane potential and molecule transports. The manuscript is written very clearly, and the microbiology part of the work appeared solid. However, there are some misconception of the physical nature of PMF and a potential issue with this part of experimental design.

The authors wrote "The membrane potential is normally created by establishing a proton (H⁺) gradient across the cell membrane", by citing a paper on mitochondria. I recommend the authors revise the biophysical basis of membrane potential in a neuroscience textbook. There was also a recent review on bacterial membrane potential dynamics. As far as I understand, in the case of *E coli*, proton gradient is not a dominant part of plasma membrane potential because there is not much proton concentration gradient (ie pH inside and outside are similar). It is worth noting that PMF consists of two components: membrane potential and proton gradient. For the measurements of membrane potential with valinomycin (Figure 2b), the tracking of the intensity change over time is essential to assess the result. Please also provide an explanation for the linear decay in the fluorescence for the experiment without valinomycin. At any rate, I am concerned about the experimental design. According to the method section, the authors incubated cells in 100mM KCl for 25min prior to the measurement. This is an extremely high concentration, which would depolarize cells. Most likely most cells are dead or damaged by the incubation in such a high KCl solution. This must be verified in order to support the authors' conclusion. In the method, the authors suggested that the addition of valinomycin induce an influx of K⁺. However, this depends on the potassium motive force. Cells would indeed likely depolarize under this condition, but it is not due to K⁺ influx. Consistent with my suspicion that cells are damaged, cells do not appear healthy in the bright field images in Fig2F. With this reason, the data presented in Figure 2 do not convincingly support the authors' conclusion, which is key for the main conclusion of the manuscript. Also, please include more details about the microscopy sample preparation as it is important to assess the results.

Minor points:

- 'rpm/min' appeared several times in the method. Rpm stands for revolution per minute and '/min' is unnecessary.
- Please specify the source of antibodies. It would be important to include the anti-PspA western blot results with *pspA* mutant to ensure the specificity of the antibody.

Reviewer #4 (Remarks to the Author):

In this manuscript, Wang et al. describe a new mechanism for starvation induced antibiotic tolerance via maintaining the proton motive force (PMF). They identify a gene related to maintaining PMF, *pspA*, as overexpressed during starvation despite cells normally reducing gene

expression in such an environment. The authors show that a *pspA* knock out has decreased survival during starvation, decreased antibiotic tolerance, and decreased cell membrane potential. They continue to show that disruption of PMF (via other methods beyond the *pspA* knock out) reduces starvation induced antibiotic tolerance. And although their results initially disagree with previously published data about the effect of PMF disruption on antibiotic tolerance by Allison et al., they determined that the discrepancy is due to timescale differences. While disruption of PMF during exponential growth can increase tolerance as previously shown, the authors demonstrated that these cells do not survive for a longer than 48 hours or through starvation after nutrient depletion. While the authors next demonstrate that loss of PMF leads to increased antibiotic uptake and reduction in efflux, they use a PMF inhibitor (CCCP) and not the *pspA* knock out. My main concern is the lack of a direct connection between this increased antibiotic uptake and decreased efflux with starvation induced upregulated gene(s) like *pspA*. I would like to see how the *pspA* knock out used early in the paper differs in antibiotic uptake/efflux. I believe that connection would strengthen this interesting work that adds to the knowledge on antibiotic tolerance and persistence.

Major

1. Did any other genes/operons involved in PMF show up in the RNA seq data? If so, including that in Table S1 would be helpful, as well as touching on this in the main text.
2. Were the experiments done in Figure 4 performed on the *pspA* deletion strain? It makes sense that CCCP is better/faster at diminishing the PMF, but if maintaining the PMF is a key component of starvation induced tolerance, it would be interesting to see how removing the ability to induce that (via the *pspA* knock out) affects antibiotic uptake/efflux directly.
3. Are there homologues of *pspA* that could be involved in PMF maintenance in the species shown in Figure 5?

Minor

1. Mention which bacterium/strain is being used in the first paragraph of results.
2. Experimental setups need to be clearer throughout the text. For example:
 - a. In the first experiment to determine gene expression changes during starvations (beginning at line 112) is there any antibiotic present?
 - b. If not, please clarify the difference in the setup of this experiment and the experiment on the *psp* knock outs (starting on line 129).
3. Explain the function of the *psp* operon when introduced (line 122).
4. What is Psp response (line 146)?
5. Line 68: "high expression level of *tolC*"  "high expression of *tolC*"
6. Line 93: lowercase *a* in *aeruginosa*
7. Some figure captions list both the definition of two and four stars for significance when only one of these is needed.
8. Fig. 2F: Why do some of the cells look like they have blunt ends and aren't nicely rod shaped?
9. There is a missing symbol showing up as a square in line 281 and multiple places throughout the supplementary materials.
10. Line 441: Tolerant cells need not be dormant.
11. Line 470: extra space before comma
12. Supplementary line 33: *rmp*  *rpm*

Responses to reviewers' comments

Reviewers' comments

Reviewer #1

Q1: Most experiments shown in the paper use ampicillin as the bactericidal antibiotic. However, it would be interesting to test other conventionally bactericidal compounds to understand whether enhanced sensitivity is observed across discrete drug classes. For example, is *pspA* sensitive to killing by ciprofloxacin in stationary phase (could be added to figure 1). Is this strain more sensitive to gentamicin or colistin? Expanding the antibiotic collection used in studies of how gene deletion mutants and PMF-dissipating chemicals impact tolerance would improve the ability to generalize some of your comments, which are limited to ampicillin currently. Furthermore, ampicillin disrupts PG biosynthesis, which one could imagine would be more synergistic with PMF disruption due to both components being at the cell envelope.

*Response: Thank you for the suggestion; we agree that it is meaningful to test other antibiotics. We have tested the effect of colistin, gentamicin and ciprofloxacin. It is hard to induce tolerance to colistin as colistin exhibits strong killing effect on starvation-induced tolerant E. coli cells, which is consistent with previous findings (PMID: 31036690, PMID: 27025620, PMID:27600051). Then we focused on gentamicin (10µg/mL) and ciprofloxacin (0.5µg/mL). When the *pspA* gene was deleted, starvation-induced bacteria cells were found to be highly sensitive to gentamicin. But deleting the *pspA* gene has little effect on ciprofloxacin tolerance (Fig. SIC, manuscript). After 144hrs of treatment, the cell density of gentamicin-treated Δ *pspA* and wild type strains was $\sim 2.5 \times 10^3$ cells/mL and $\sim 1.5 \times 10^4$ cells/mL, respectively, whereas that of ciprofloxacin-treated Δ *pspA* and wild type was $\sim 9.8 \times 10^5$ cells/mL and $\sim 1.2 \times 10^6$ cells/mL, respectively. Consistent to this result, PMF-dissipating chemical CCCP(1µM) renders the cells more sensitive to gentamicin than ciprofloxacin in both wide type and Δ *pspA* strains (Fig. S3B-E, manuscript). Hence was conclude that dissipation of PMF suppresses tolerance to gentamycin but not ciprofloxacin.*

Q2: The Western blot in figure 1b needs to be re-conducted and re-quantified since the loading control GAPDH is clearly in higher abundance in starvation than in log phase.

Response: Thank you for pointing out this issue. We re-conducted western blot and the result is shown as Fig. 1B in the revised manuscript. The new figure clearly shows that expression of PspA is strongly up-regulated during nutrient starvation.

Q3: In figure 2, the authors use valinomycin as a PMF-disrupting molecule, showing that the *pspA* deletion strain displays higher DiSC3(5) fluorescence upon exposure to valinomycin. This molecule is a specific disruptor of the delta psi component of the PMF. How does the *pspA* strain respond to treatment with nigericin for example, which is a specific disruptor of the delta pH component of the PMF? Is the *pspA* strain equally sensitive to disruption of both components of the PMF, or just the delta psi?

Response: We use BCECF-AM (Invitrogen, B1150) to test the change of ΔpH . 525/610 nm fluorescence emission ratios (excited at 488 nm) are used to measure intracellular pH (PMID: 8672290). According to the result, the PC5.5 VS FITC ratio has little change between wild type (WT) and $\Delta pspA$ ($pspA$) under starvation (Fig. a1&a3). The ratio is similar between nigericin-treated starved wild type and $\Delta pspA$ (Fig. a2&a4). So the effect of $\Delta pspA$ on tolerance is due to the change of delta psi. We searched the publications which reported the effect of PMF on bacteria tolerance and found that most of publications reports that membrane potential affect tolerance and few was found on the links between ΔpH and tolerance (PMID: 26051177, PMID: 31327636).

Fig.(a) The profile of 525nm emission and 610nm emission were similar between wide type (a1) and $\Delta pspA$ (a3), with nigericin treated group (a2&a4) as positive control in which the fluorescence signal significantly shifted comparing with the non-nigericin treatment group. The bacteria cells were identified by FSC and SSC parameters.

Q4: Does valinomycin or nigericin potentiate antibiotics in deep stationary phase, like what you observe with CCCP or (to some extent) NaN₃?

Response: We tested the effect of valinomycin (100 μ M) and nigericin (100 μ M) on tolerant bacterial cells and found that they cannot potentiate the killing effect of antibiotics (fig. b1&b2). In Fig 2 which shows the results of membrane potential measurement, KCl was added before treatment with valinomycin to form a potassium ions-based motive force. When valinomycin was combined with ampicillin treatment, we did not add KCl because adding ions is not a standard routine to test synergistic effect of two drugs.

Fig.(b) The survival population after treatment with valinomycin /nigericin or combinations of ampicillin in 24hrs starvation- induced wide type (b1) or Δ pspA (b2). The result shown valinomycin/nigericin did not potentiate ampicillin.

Q5: Comparing figure 2a and figure 2b, it appears as if in this case the pspA knockout on its own (figure 2b) does not have a large fitness disadvantage relative to the wild type strain on its own (figure 2a). However, there is a clear fitness cost that we can observe in figure 1. Can the authors explain this discrepancy?

Response: In Fig 2B, the cells are in exponential phase with sufficient nutrient (LB broth cultured). The membrane potential of wild type and Δ pspA cells as detected by DiSC₃(5) is similar. But under starvation, membrane is depolarized in Δ pspA when comparing with wild type (Fig 2C, manuscript), indicating the cells cannot maintain membrane potential under starvation stress if pspA is deleted. Fig 1 was tested under starvation to show the fitness cost, so Δ pspA strain has fitness disadvantage under starvation but not in the exponential phase, presumably due to the fact that PMF can be produced by oxidative phosphorylation and need not be maintained by PspA. This is also consistent with our observation that pspA expression level in exponentially growing cells of the wild type strain is extremely low. Fig 2A and Fig 2B depict different membrane characters. In Fig 2A, SYTOX Green was used to assess membrane permeability. Colistin was used in the positive control as it destroys membrane integrity. The results show that membrane permeability is not changed in Δ pspA strain during starvation.

Q6: In figure 5, there is no evidence that combining CCCP with ampicillin is synergistic. In all cases except *P. aeruginosa*, CCCP is completely bactericidal on its own, and the CCCP + ampicillin curve simply overlays with the CCCP curve. This is unconvincing to show that PMF maintenance is important for antibiotic tolerance across bacterial species. This experiment needs to be optimized to clearly show that chemically or genetically modulating PMF can enhance bactericidal antibiotic efficacy against tolerant cells across different bacterial species.

Response: We speculate that the reason why cells are completely killed is that the concentration of CCCP at 100 μ M is too high and exhibited bactericidal effect by itself. We then tested different

concentrations of CCCP. CCCP potentiated ampicillin killing efficacy against starvation-induced tolerant cells without showing significant toxicity by itself at 1 μ M in *E. coli*, 10 μ M or 5 μ M in *S. typhimurium*, 1 μ M or 0.1 μ M in *A. baumannii*, 1 μ M in *S. aureus* and 50 μ M in *K. pneumoniae* respectively (Fig. c1-c5). Some of these results were included in the revised manuscript (Fig. 5, manuscript).

Fig.(c) Changes in the size of antibiotic-tolerant sub-population in *E. coli* (c1), *S. typhimurium* (c2), *A. baumannii* (c3), *S. aureus* (c4) and *K. pneumoniae* (c5) which had been starved for 24hrs, followed by treatment with 10 \times MIC ampicillin (AMP) alone, CCCP alone and CCCP in the presence of 10 \times MIC ampicillin. CCCP(100), 100 μ M CCCP; CCCP(50), 50 μ M CCCP; CCCP(10), 10 μ M CCCP; CCCP(5), 5 μ M CCCP; CCCP(1), 1 μ M CCCP; CCCP(0.1), 0.1 μ M CCCP.

Q7: 100 ug/ml ampicillin is a fairly high concentration to be working with. Is ampicillin activity

not observed below this concentration in any of the genetic or chemical conditions? I suggest providing some dose-response data showing at which concentration (at some endpoint, 144 hrs perhaps) ampicillin potentiation is observed in the various genetic and chemical conditions.

*Response: Tolerance is a phenomenon in which bacteria can survive lethal dosage of antibiotic treatment and can re-grow after antibiotic withdrawal. In order to obtain ampicillin-tolerant population, treatment with high concentration ampicillin is necessary, only those which survive such high dosage are regarded as tolerant. In previous studies, 125 μ g/ml ampicillin was used to treat *E. coli* K-12 MG 1655 ($MIC_{amp}=8\mu$ g/ml) (PMID:32723793); 100 μ g/ml ampicillin was used in *E. coli* K-12 MG 1655 in Kim Lewis's report (PMID: 15576765). *E. coli* K-12 BW 25113 (we use) is genetically similar to MG 1655 and MIC of ampicillin is 8 μ g/ml for both strains. We therefore chose 100 μ g/ml ampicillin for tolerance assay. We then tried the combination of CCCP with 100 μ g/ml, 50 μ g/ml and 10 μ g/ml ampicillin and the result shown that the synergistic effect decreased along with the decreasing concentration of ampicillin (Fig. d). Hence 100 μ g/ml is the proper concentration to investigate the potentiation effect of various chemical / gene deletion.*

Fig. (d) The survival population after 144hrs treatment with different concentrations of ampicillin in the presence or absence of CCCP (1 μ M) upon 24hrs starved wide type. Significant decrease of population in the presence of CCCP was observed only when the concentration of ampicillin reached 100 μ g/ml.

Q8: Does a *pspA tolC* double deletion display enhanced sensitivity to ampicillin relative to either single-gene deletion strain, or does it simply look like the *pspA* strain, since presumably this would be the gene that would have more impact on *tolC* rather than vice versa.

*Response: The size of the tolerant population of Δ pspA, Δ tolC and Δ pspA*tolC* that survived 144hrs of ampicillin treatment was $\sim 3 \times 10^5$ CFU/mL, $\sim 5 \times 10^4$ CFU/mL and $\sim 2 \times 10^3$ CFU/mL, respectively (Fig. e). The double deletion strain exhibited enhanced sensitivity to ampicillin.*

TolC is a component of many efflux pumps so the functions of a number of efflux pumps will be inhibited by deleting *tolC*, hence the size of the tolerant population also decreased significantly when *tolC* was deleted. The tolerance level was further reduced when both *pspA* and *tolC* were deleted, as both PMF and efflux activities were affected. Apart from efflux pumps, PMF also affected other membrane functions, hence double deletion would lead to much lower survival in tolerant cells.

Fig. (e) The survival population at indicated ampicillin treatment time point upon 24hrs starved cells of wild type, $\Delta pspA$, $\Delta tolC$ and $\Delta pspA/tolC$, with no ampicillin-treatment group as control.

Reviewer #2

Q1: The authors should quantify ATP in their cells (e.g., using Promega's BacTiter Glo kit) to ensure that while PMF in some of their samples are dissipated (valinomycin, CCCP, sodium azide-treated samples), the cells are retaining enough adenylate charge to remain viable (Chapman et al., 1971, PMID: 4333317).

Response: We use Promega's Ba Titre Glo kit (G8230) to test valinomycin (100 μ M), CCCP (1 μ M), sodium-azide (5mM) treated cells for 144hrs and also do viable cell counts of these samples. We followed PMID:4333317 to test adenylate energy charge and found the charge is above 0.5, indicating cells maintaining viability (Table A). The reviewer is concerned whether a change in ATP level is associated with cell viability and growth, the CFU recorded show that cell viability and growth is similar after treatment with CCCP and valinomycin; and decreased slightly after sodium azide treatment. CFU of untreated wild type and sodium azide-treated wild type cells are $\sim 2.5 \times 10^7$ CFU/mL and $\sim 6.6 \times 10^6$ CFU/mL, respectively. CFU of untreated $\Delta pspA$ and sodium azide treated $\Delta pspA$ are $\sim 7.4 \times 10^6$ CFU/mL and $\sim 4.0 \times 10^5$ CFU/mL, respectively (Fig. f). The CCCP concentration used (1 μ M) is consistent with that suggested by Reviewer 1, Q6. CCCP concentration in Fig 3 of the revised manuscript will also be changed to 1 μ M.

Table A. Adenylate energy charge values after compounds treatment

Strain	adenylate energy charge			
	Saline	NaN ₃	CCCP	Valinomycin
wild type	0.62±0.03	0.56±0.03	0.55±0.03	0.60±0.02
Δ pspA	0.61±0.04	0.54±0.02	0.55±0.02	0.62±0.04

Fig. (f) The survival population after sodium azide, CCCP and valinomycin treated for 144hrs upon 24hrs starvation-induced wild type/Δ*pspA* cells. CFUs of wild type cells were $\sim 2.5 \times 10^7$ cells/mL, $\sim 6.6 \times 10^6$ cells/mL, $\sim 3.9 \times 10^7$ cells/mL and $\sim 2.8 \times 10^7$ cells/mL after saline, sodium azide, CCCP and valinomycin treatment respectively. CFUs of Δ*pspA* cells were $\sim 7.4 \times 10^6$ cells/mL, $\sim 4.0 \times 10^5$ cells/mL, $\sim 8.3 \times 10^6$ cells/mL and $\sim 6.4 \times 10^6$ cells/mL after saline, sodium azide, CCCP and valinomycin treatment respectively.

Q2: In their RNA-Seq and antibiotic tolerance experiments, the authors inoculate *E. coli* cultures that had reached OD600 of 0.2 in saline (0.85% or 0.9% NaCl). As phage shock proteins (e.g., *pspA*) can be induced by prolonged stationary phase at high pH and osmotic shock, could the choice of “starvation” media contribute to the increased expression of *pspABCD*, as detected in their RNA-seq experiments? Why did the authors choose to use NaCl solutions as opposed to phosphate-buffered saline or defined media lacking a carbon source (e.g., M9 salts without glucose) as their starvation condition? It is unclear whether the starvation transcriptome that the authors report here are generalizable to the transcriptomes of cell starved under buffered conditions.

Response: We followed Ruchira et al., 2019, PMID: 31653977 who used saline to create a nutrient deprivation condition. 250mM NaCl (14%) was used to create salt stress in *Desulfovibrio vulgaris* and the concentration we use is 9% NaCl (Zhou et al., 2017, PMID: 29138306). As we mainly focus on the effect of *pspA*. We did RT-PCR of gene *pspA* under saline and PBS for 24hrs (log phase in LB media as negative control), which is consistent with the culture condition of RNA-Seq. *pspA* was found to be over-expressed both in saline and PBS (about 100 times in saline and 25 times in PBS) (Fig. g). The degree of over-expression is higher

in saline, but the expression level was also enhanced significantly in PBS.

Fig. (g) The relative transcription level of *pspA* gene was quantified by RT-PCR. Log phase cells was used as negative control with 16s RNA segment as endogenous control.

Q3: The authors primarily focused on the overexpression of *psp* genes in their RNA-seq data, yet they do not comment on other up-regulated genes identified in their analysis. For instance, a few transporters and efflux pumps are as highly expressed as *pspABC*.

*Response: We also agree with the reviewer that other up-regulated genes are also very important and needs further exploration. Actually, we have examined the relationship between bacteria tolerance level and the over-expressed transporter genes by assessing the tolerance level of specific gene deletion mutants. We identified transporters affect tolerance formation, such as *emrKY-tolC* pump. We have prepared a separated manuscript to report these findings. The *pspABCDEF* genes were all over-expressed under starvation. Since a large number of genes were over-expressed, we only selected genes in which all members of the family were over-expressed for further study. In this work, we focused on genes in the *psp* gene family.*

Q4: The authors did not report on the genes that were down-regulated in their RNA-seq experiments.

Response: When bacteria undergo starvation, most of the cells will enter into dormancy and reduce unnecessary metabolism to enhance survival fitness (Kim.et al, PMID: 17143318). Under this condition, most of genes are down-regulated and the over-expressed genes are expected to be involved in mediating tolerance formation. The purpose of this work is to identify and investigate active cellular mechanisms involved in tolerance formation and hence the up-regulated genes; the scope of our work does not cover the down-regulated genes at this stage.

Q5: Fig. 1A: The authors did not report their CFU counts for t=0.

Response: We have included the CFU counts at t=0 in the revised figures. The CFU of wide type at t=0 is $\sim 7.2 \times 10^7$ cell/mL, which is similar to that at t=4hrs ($\sim 7.0 \times 10^7$ cell/mL). The CFU of Δ pspA at t=0 and t=4 is $\sim 7.2 \times 10^7$ cell/mL and $\sim 7.0 \times 10^7$ cell/mL respectively. CFU at t=0 has been included in the reversed manuscript.

Q6: The reduced culturability of the Δ pspA mutants by ~ 4 -fold after 144 h of starvation could have contributed to the reduction in ampicillin persisters throughout the course of treatment. Are the differences in survival between WT and Δ pspA cells without treatment statistically different? How about after treatment? As Δ pspA did not eradicate the population during starvation with and without ampicillin treatment, the title of Fig. 1 indicating that the Psp response is “essential” for survival and tolerance, is misleading.

Response: At 144hrs, the CFU of Δ pspA (without ampicillin treatment) is $\sim 7 \times 10^6$ cells/mL and CFU of wild type (without ampicillin treatment) is $\sim 2.5 \times 10^7$ cells/mL, P values of these two groups is 0.002 (unpaired, two-sided Student's test). At 144hrs, the CFU of Δ pspA (with ampicillin treatment) is $\sim 3 \times 10^5$ cells/mL and CFU of wide type (with ampicillin treatment) is $\sim 4 \times 10^6$ cells/mL, survival rate after ampicillin treatment in Δ pspA and wide type are $\sim 4.3\%$ and $\sim 16\%$, respectively. Cells become less tolerant to ampicillin and also less adaptable to starvation stress upon deletion of pspA. Psp response elicited during nutrient starvation affects bacterial survival and antibiotic tolerance. We have modified the title of Fig.1 and tone down our description on the role of PspA in maintaining phenotypic antibiotic tolerance.

Q7: The authors should complement their deletion mutants.

Response: We complement the pspA, tolC, nuoI and ndh genes back into deletion mutants by inserting target genes into pBAD18 plasmids (resistant to kanamycin or chloramphenicol), followed by transformation of the plasmid into the deletion mutants. The tolerance level of the complemented strains is similar to that of wild type (Fig. S1B, S3F, S4G, manuscript).

Q8: Fig. 1B: The authors should have tried to load similar amounts of proteins in each lane of the gel (the GAPDH band looks to be a few folds fainter in the Log sample relative to the Starvation sample), and they should show the full blot in the supplemental figures. The dark background observed in the GAPDH bands can potentially affect quantification using ImageJ.

Response: We re-conducted western blot and the result as shown in Fig. 1B (manuscript) confirms that, when compared to exponentially growing cells, PspA is expressed in a much larger amount during starvation. The blot figure is the original image we obtained from the Azure c600 Gel Imaging System. We only did some cuttings to remove excessive white background. ImageJ was only used to calculate the relative protein level.

Q9: Fig. 2C: Perhaps the authors should add DiSC3(5) at the time of incubating the cells in saline and quantify the change in fluorescence in the population at different times thereafter. The way the experiment is set up now, it is unknown whether the fluorescence is higher in the Δ pspA mutants because of decreased dye uptake or increased dye release and dequenching.

Response: The timeframe for stabilizing DiSC₃(5) uptaking process is quite short (within 5mins) and its import and export depends on membrane potential (Winkel et al., PMID: 27148531, Fig 2A and Fig 4A; Morin et al., PMID: 21189348, Fig 5A). We incubated DiSC₃(5) with cells for 25mins before measurement. We resuspended cells in saline for 24hrs to impose starvation stress for 24hrs, if DiSC₃(5) was added at the time of resuspending the cells in saline, DiSC₃(5) have to be detectable for 24hrs. It would be too long and no signal could be detected as the dye would have been completely quenched after the 24hrs incubation.

Q10: How were the relative fluorescence values determined? Why are the values lower for the log phase samples in Fig. 2D lower than those in Fig. 2B and the starvation samples lower in Fig. 2D compared with those in Fig. 2C.

Response: This is because we used different microplate readers, Fig. 2B and 2C show results produced by Clariostar Microplate Reader (BMG LABTECH) in the Hong Kong Polytechnic University and Fig. 2D was results produce by SpectraMax iD3 Multi-Mode Microplate Readers (Molecular Devices) after we have moved to City University of Hong Kong. It is due to different internal calibration of the instrument itself. We used the BMG Microplate Reader to do calibration and generate a new Fig. 2D (manuscript), in which the fluorescence intensity is similar in log phase cells and starvation cells; the intensity value is now consistent with Fig. 2B,2C.

Q11: The microscopy images in Fig. 2E and 2F do not agree with the data from Fig. 2C-D. According to Fig. 2C, fluorescence of the DiSC₃(5) dye is quenched in WT cells, yet these cells were the brightest in the microscopy experiments. On the other hand, the valinomycin-treated samples were the most fluorescent when they were quantified using a plate reader, yet these populations were dark in the microscopy images. If, as the authors state in line 209, the “membrane potential is too low” in the Δ pspA mutants, how did the dye enter these mutants in Fig. 2C?

Response: In this whole microplate reading experiment, DiSC₃(5) was not washed and the recorded signals represent overall fluorescence signals (intracellular and extracellular), while in microscopy, only intracellular fluorescence signal was detected. When adding DiSC₃(5) into cell culture, it will enter the cells due to existing cell membrane potential, this process will cause the fluorescence signal to rapidly decrease as signal in intracellular DiSC₃(5) is strongly quenched. After 3~5 mins of stabilization, ionophore (for example, valinomycin which is K⁺ carrier) was added to cause membrane depolarization. DiSC₃(5) is sensitive to the change in membrane potential and are released from cell to the medium, resulting in increase in fluorescence signal. This is due to that the fact that the released DiSC₃(5) will be de-quenched (Winkel et al., PMID: 27148531, Fig. 2A, Fig. 4A and Fig. 5). Therefore, when the membrane was depolarized, the overall fluorescence signal increased (Fig. 2B, C, D, manuscript) but intracellular fluorescence signal decreased (Fig. 2F, manuscript). For example, in Strahl et al., PMID: 20566861 Fig. S6B, in the presence of 300 mM KCl, valinomycin supplementation caused depolarization and fluorescent image is dark. Associated with the response to Q9, we

added DiSC₃(5), stabilize for 25mins (sufficient for DiSC₃(5) to penetrate into cell) and then add valinomycin to cause depolarization (100mM KCl was supplemented). When membrane is depolarizing, intracellular DiSC₃(5) get out into medium causing overall fluorescent signal to increase as the out-going DiSC₃(5) molecules are de-quenched. However, the intracellular (not overall) fluorescence signal decreases when membrane is depolarized as intracellular DiSC₃(5) get out of the cells.

Q12: Lines 244-246: Perhaps it is not surprising that starving bacteria maintain basal PMF, otherwise they would be deenergized and dead.

Response: Yes. We also agree that starving bacteria maintain basal PMF. The basal PMF not only guarantee viability but also plays a role in maintaining phenotypic tolerance. Our data indicate that sodium azide and CCCP, which dissipate PMF, strongly affect tolerance maintenance (Fig. 3A&3B, manuscript). We have amended this sentence.

Q13: Line 252: I disagree that the Δ pspA mutants “lacks the ability to maintain PMF” as the authors claim, as many of these cells remained viable after 144 h in saline.

Response: Thanks. PspA plays a role in maintaining PMF rather than that lack of PspA means total lack of ability to maintain PMF. We have rephrased the sentence.

Q14: Fig. 3E and F: The authors should include WT cells treated with antibiotics (AMP/ GEN) alone as controls for comparison.

Response: In Fig. 3E and F, exponential bacterial cells were induced to form tolerant cells by CCCP (PMID: 30123191). So cells are still alive after long term ampicillin or gentamicin treatment (at least 72hrs). Fig. 3E&F indicated that tolerant cells could not maintain tolerance to gentamicin and was killed by gentamicin in the presence of CCCP. But with ampicillin or gentamicin alone treatment, exponential cells are totally eradicated after 4hrs as the cells are susceptible to antibiotics without tolerance induction by CCCP. To test whether tolerant cells become susceptible and can be killed by antibiotics again in the presence of CCCP, we treat starvation-inducing cells with ampicillin and gentamicin alone and compared them with CCCP/ antibiotic synergetic treatment (Fig. 3A,3B, S3B, S3C; Reviewer1, Q1). The size of the surviving population decreased more rapidly after treatment with CCCP and antibiotic (ampicillin or gentamycin). These results indicated that dissipation of PMF actually negatively affects tolerance against ampicillin and gentamicin in the long term (144hrs). We replaced the former Fig. 3E and F with figures which is Fig. S3B&C in the revised version.

Q15: Line 300: I also disagree with the authors' claims that their findings challenge the existing view of the impact of PMF suppression on dormancy and tolerance or previous findings that addition of glycolytic intermediates can enhance aminoglycoside-mediated killing. As the authors stated, they are treating E. coli with CCCP for a much longer duration than in previous studies. Furthermore, the authors never quantified ATP/ adenylate charge to ensure that their cells retained sufficient ATP/ adenylate charge to remain viable.

Response: We will delete those sentences to avoid confusion. We are looking at a much longer term and show that PMF is actually essential for survival of tolerant cells in the long term. There is no contradiction to the previous findings. The adenylate charge was presented in Q1.

Q16: Fig. 4: As the authors are looking at BOCILLIN uptake in a phenotypically heterogeneous population, it would have been better for them to quantify fluorescent cells by flow cytometry rather than using a plate reader.

Response: In the former manuscript, we tested BOCILLIN uptake by confocal microscopy. We quantified BOCILLIN fluorescence intensity by flow cytometry (Fig. 4A-4D, S4B-S4F, manuscript). Water was used as negative control to confirm the FSC/SSC values of bacterial cells and P1 was set as the targeted cells. The fluorescence intensity which is higher than 10^3 (P2) is 10.45% in the no-CCCP group, while P2 is 97.86% in the CCCP-treated group. This finding is consistent with that of confocal microscopy.

Q17: Fig. 4 A&B: If the fluorescence of the “No CCCP” sample is 2-fold lower than the “CCCP” sample, why is the fluorescence of the “no CCCP” control not detected at all in the microscopy images?

Response: When quantifying the fluorescence of two groups, we selected 30 cells to record the average fluorescence level as there are too many cells in the observation field for measurement of fluorescence. This method will cause huge error range and thanks the reviewer for suggesting flow cytometry quantification (Q16). We have replaced Fig. 4A&B with the flow cytometry result to avoid this problem. In the flow cytometry result, the difference between the fluorescence intensity of the two groups could be compared by quantifying the percentage of P2.

Q18: Fig. 4 C&D: Is the Nile Red accumulation data and the survival of the mutants statistically different from the controls (no CCCP and WT controls)?

Response: Yes. For Fig. 4C (Fig. 4E, revised manuscript), at the endpoint (Time=30mins), The RFU of “no CCCP” group is 6390,6350,6595,6289,6709 and the RFU of “with CCCP” group is 11503,10220,9548,9323,9513. P value of these two groups is 0.0007(two tailed Student’s test). For Fig. 4D (Fig. 4F, revised manuscript), the CFU/mL of ‘ Δ tolC’ group at time=144hrs is 1.8×10^7 , 2.1×10^7 , 2.5×10^7 and the CFU/mL of ‘ Δ tolC +Amp’ group at time=144hrs is 4.8×10^4 , 5.1×10^4 , 5.2×10^4 . P value of these two groups is 0.008(two tailed Student’s test)

Q19: Fig. 5: What is the MIC of ampicillin against Salmonella? Why would the authors test P. aeruginosa, K. pneumoniae, and A. baumannii when these species are naturally resistant to AMP? The data in panels B-F mainly shows that CCCP can lead to death of these organisms.

Response: MIC of ampicillin is 1 μ g/ml for Salmonella strain PY01. We understand that

ampicillin is not active against the test bacterial species, such as P. aeruginosa. This may be due to various factors such as the ability to undergo strong efflux to expel antibiotics out of the cell. We reasoned that if PMF is important for survival of bacteria under starvation stress, the PMF-dissipating agent CCCP would also reduce the survival fitness of these organisms. Consistent with our speculation, high concentration CCCP alone eradicates the starved cells. Apart from that, our paper mainly studies the effect of PMF on prolonged tolerant cells. We combine different concentration of CCCP with 10×MIC ampicillin to treat starvation-induced tolerant cells in different species. The results show that low concentration of CCCP enables ampicillin to kill those tolerant cells again without showing toxicity as the survival population of only CCCP treatment group is similar to that of non-treatment group (Reviewer1, Q6, Fig. c1-c5). Some of these results were included in the revised manuscript (Fig.5A-5E, manuscript).

Additional Comments

AQ1: Overall, the correct terminology should be “dissipation of PMF” and “depolarization of the membrane”. “PMF depolarization”, as the authors stated in line 85, is incorrect.

Response: Thanks for pointing out our mistakes; we have corrected these errors in the revised manuscript.

AQ2: Lines 84-87: In the description of the work by Verstraeten and colleagues, “yet in a follow-up study” should be replaced with “and in a follow-up study”, because if membrane depolarization leads to dormancy, it makes sense that repolarization contributes to persister awakening.

Response: We have replaced those words with “and in a follow-up study” in the reversed version.

AQ3: On that note, Ref. 19 is incorrect, as the authors should be citing the works by Verstraeten and colleagues.

Response: We are sorry for the mistake, it has been corrected in the reversed manuscript.

AQ4: Lines 95-96: It should be noted that the compounds tested in Refs. 17 and 18 do not only dissipate PMF, they can permeabilize the membrane (glycopolymers) or inhibit ATP synthase (diarylquinolines). As such PMF dissipation is not their only mechanism of killing.

Response: Yes. Apart from dissipating PMF, PAAG (glycopolymers) permeabilizes membrane and diarylquinolines inhibit ATP synthase. Even though there is no evidence that their killing effect is merely due to PMF dissipation, PMF is still a potential target for screening compounds against tolerant bacterial cells. Farha et al., used PMF as a target to screen compounds against resistant S. aureus though the compounds they screened out were found to cause decrease in ATP level, too (PMID: 23972939).

AQ5: Why did the authors choose to use 0.9% NaCl for their tolerance assay, but used 0.85% NaCl for their RNA-seq experiments?

Response: In this study, 0.9% NaCl was used, including RNA-Seq. the discrepancy has been corrected.

AQ6: As all of the PMF disruption methods that the authors used also disrupts mitochondrial PMF, their claim at the end of their abstract that disrupting bacterial PMF is a potential strategy for treating chronic infection may not actually be feasible.

Response: Actually, our lab has already identified an FDA approved drug which disrupts bacterial PMF and can act as an adjuvant to eradicate tolerant cells. Animal experiments show that this drug has little toxicity. This finding has been submitted to another journal for reviewing. Most of PMF-dissipating chemicals exhibited high cytotoxicity, such as CCCP and sodium azide. But it was reported that compounds such as PAAG and AM-0016 killed tolerant bacteria by disrupting PMF without exhibiting toxicity (PMID: 30123191, PMID: 23441632, PMID: 27158932). The compounds reported by Farha et al. also exhibited acceptable cytotoxicity (PMID: 23972939).

AQ7: Additional details on their RNA-seq sample preparation is warranted. What was the depth of read? Was rRNA depletion carried out during library preparation?

Response: Yes. We have described in the revised manuscript that rRNA depletion was carried out using illumina Ribo-Zero Plus rRNA Depletion Kit. The total reads were about 3 million. The median of RPKM was about 58. We only analyzed the genes whose RPKM was above 5 in both log phase and starvation group to guarantee that RNA-seq data of the genes were statistically meaningful. We have also described the method of RNA-Seq in details in the reversed version.

Typographical errors

TE1: Title: (“Active Maintenance of Proton Motive Force is Key Mechanism of Starvation-induced Bacterial Antibiotic Tolerance” should be changed to “Active Maintenance of Proton Motive Force is a Key Mechanism of Starvation-induced Bacterial Antibiotic Tolerance”).

Response: We have made the correction in the revised version.

TE2: Line 93: Replace “Pseudomonas Aeruginosa” with “Pseudomonas aeruginosa”.

Response: Corrected accordingly.

TE3: Fig. 2B-D: The symbols are really difficult to discern. The authors should consider using larger symbols and using filled in vs. hollow symbols to compare their +/- valinomycin symbols.

Response: Thanks for pointing out this problem. We have used larger symbols and distinguished +/- valinomycin by filled in and hollow symbols respectively.

TE4: Line 281: Do the authors mean to say NADH dehydrogenase II? The symbol is replaced with a square in the document.

Response: Yes. It's NADH dehydrogenase II, this bug was formed during converting word format to PDF format. We have amended the error.

TE5: Line 291: I am not sure that the authors are correct in saying that $\Delta ndh \Delta nuoI$ resulted in the “strongest tolerance suppression effect recorded”. Deletion of some of the SOS response genes (e.g., *recA* and *recB*) results in a similar magnitude of decrease in persistence toward fluoroquinolones, even in non-growing populations.

Response: We will tone down and amended the wording accordingly.

TE6: Line 29: “Pkd46” should be changed to “pKD46”.

Response: We have made this change in the revised manuscript.

TE7: Line 33: “rmp” should be changed to “rpm”.

Response: We have corrected this error in the revised manuscript.

TE8: On line 36, 44 and other locations: “oC” is not visible and a square is present instead.

Response: Sorry about that. The error has been corrected in the revised manuscript.

TE9: Line 45: “OD600” should be modified to “OD₆₀₀”.

Response: It has been modified to “OD₆₀₀” in the revised manuscript.

TE10: Line 79: Please remove the comma at the end of the sentence.

Response: We have made this change in the revised manuscript.

TE11: Line 206: “DiSC3(5)” should be changed to “DiSC₃(5)”.

Response: It has been modified to “DiSC₃(5)”.

Reviewer #3

Q1: The authors wrote “The membrane potential is normally created by establishing a proton (H⁺) gradient across the cell membrane”, by citing a paper on mitochondria. I recommend the authors revise the biophysical basis of membrane potential in a neuroscience textbook. There

was also a recent review on bacterial membrane potential dynamics. As far as I understand, in the case of E coli, proton gradient is not a dominant part of plasma membrane potential because there is not much proton concentration gradient (ie pH inside and outside are similar). It is worth noting that PMF consists of two components: membrane potential and proton gradient.

Response: Thank you for pointing this out. We also agree that PMF consists of membrane potential and proton gradient. When studying the effect of PMF on tolerant bacteria cells, we referred to Allison et al., 2016, PMID: 21562562 published in Nature, in which the authors used carbocyanines dye to test PMF change in tolerant bacteria. We tested the effect of nigericin which facilitates an electroneutral exchange of H⁺ and K⁺ ions with the effect of depleting the ΔpH. Nigericin cannot re-sensitize tolerant cells to antibiotic (Fig. h1&h2). When studying the effect of PMF on bacteria tolerance, most papers report that membrane potential affect tolerance and few report links between ΔpH and tolerance (PMID: 21562562, PMID: 26051177, PMID: 31327636).

Fig. (h) The survival population after Nigericin (100μM) alone or combining with ampicillin (100μg/mL) treated for indicated times upon starved 24hrs cells of wild type (h1) or ΔpspA (h2), with non-treatment and only ampicillin treatment as control. CFUs of only nigericin treatment group was similar to that of non-treatment group, indicating nigericin itself did not kill tolerant cells. CFUs of combining treatment group was similar to that of only ampicillin treatment, indicating nigericin did not re-sensitize tolerant cells to become susceptible to ampicillin again.

Q2: For the measurements of membrane potential with valinomycin (Figure 2b), the tracking of the intensity change over time is essential to assess the result. Please also provide an explanation for the linear decay in the fluorescence for the experiment without valinomycin. At any rate, I am concerned about the experimental design. According to the method section, the authors incubated cells in 100mM KCl for 25min prior to the measurement. This is an extremely high concentration, which would depolarize cells. Most likely most cells are dead or damaged by the incubation in such a high KCl solution. This must be verified in order to support the authors' conclusion.

Response: Line decay is due to the fact that DiSC₃(5) can bind on the polystyrene-surface of the microtiter plate (Winkel et al., PMID: 27148531, Fig. 3A). This decay can be halted by adding BSA, but it cannot be stopped completely. The population size recorded after 100mM KCl treatment for 24hrs is $\sim 7 \times 10^7$ CFU/mL and the population without KCl treatment is $\sim 9 \times 10^7$ CFU/mL (Fig. i). The size shrunk slightly after 100mM KCl treatment. As valinomycin is K⁺ carrier, supplementation of KCl is required to make valinomycin transport K⁺ into the cell and the depolarization caused by valinomycin only took place in the presence of sufficient K⁺ in the medium (Winkel et al., PMID: 27148531, Table 2, Strahl et al., PMID: 20566861 Fig. S6B, 300mM KCl was used).

Fig. (i) The survival population after 100mM KCl treatment for 24hrs with non-treatment as negative control. CFUs were $\sim 8.5 \times 10^7$ cells/mL and $\sim 8.4 \times 10^7$ cells/mL in the absence and presence of KCl respectively.

Q3: In the method, the authors suggested that the addition of valinomycin induce an influx of K⁺. However, this depends on the potassium motive force. Cells would indeed likely depolarize under this condition, but it is not due to K⁺ influx.

Response: Valinomycin acts as potassium ionophore and can transport potassium across the membrane driven by potassium motive force. We used valinomycin in the presence of 100mM KCl in the media as the positive control (depolarizing group). According to Strahl, et al. PMID: 20566861, Fig. S6B, when the medium was supplemented with KCl, valinomycin would transport K⁺ into cells and caused depolarization in B. subtilis. In E. coli, the concentration of KCl used in valinomycin-treated membrane potential test is 100mM (Morin, et al. PMID: 21189348).

Q4: Consistent with my suspicion that cells are damaged, cells do not appear healthy in the bright field images in Fig2F. With this reason, the data presented in Figure 2 do not convincingly support the authors' conclusion, which is key for the main conclusion of the manuscript.

Response: We compared the bright field of 100mM KCl treated cells (Fig. 2E, WT, revised

manuscript) with non-KCl treated cells (Fig. i, WT, Reviewer4, Q2) and found the morphology and size of the cells in these two groups were similar (Fig. j). We add 2 μ L stained cells to the slide and press the slide to make the cell culture layer thin in order to achieve better vision. We observed that cells would display several overlapping layers if we didn't press slide. The pression would make the cultures dry more quickly. To ensure consistency of the experiments, we usually prepare four samples at the same time (WT, WT+V, Δ pspA, Δ pspA+V), which require long observation time. Due to extension of the observation time, samples may become dry and the morphology of the cells might be slightly affected.

Fig. (j) The bright field images of KCl-treated and non-KCl treated wild type cells observed by confocal. The shape and size of the cells were identical between the two groups (scale bar: 4 μ m).

Q5: Also, please include more details about the microscopy sample preparation as it is important to assess the results.

Response: We have included the details on microscopy sample preparation in the revised manuscript.

Minor

MQ1: 'rpm/min' appeared several times in the method. Rpm stands for revolution per minute and '/min' is unnecessary.

Response: Thank you. We have corrected this mistake.

MQ2: Please specify the source of antibodies. It would be important to include the anti-PspA western blot results with pspA mutant to ensure the specificity of the antibody.

Response: Anti-PspA antibody is polyclonal and from rabbit. The result of western blot

confirmed that PspA was not produced in Δ pspA (Fig. k).

Fig. (k) Western blot of the PspA protein in wild type and Δ pspA at cell density of OD_{600} 0.2 (exponential phase), with the endogenous protein GAPDH as control.

Reviewer #4

Q1: Did any other genes/operons involved in PMF show up in the RNA seq data? If so, including that in Table S1 would be helpful, as well as touching on this in the main text.

Response: We checked other envelope stress response genes. Rcs response was also reported to maintain membrane potential (PMID: 28588134), rcsA was up-regulated approximately 100 times, but rcsD was down-regulated about 3 times and the expression level rcsB and rcsC as similar before and after starvation. The cpxP gene was up-regulated 268 times. CpxQ which is the 3' un-transcribed region of cpxP transcript, was reported to be involved in preserving PMF (PMID: 26805574, PMID: 27048800). We have included the information in the revised manuscript.

Q2: Were the experiments done in Figure 4 performed on the pspA deletion strain? It makes sense that CCCP is better/faster at diminishing the PMF, but if maintaining the PMF is a key component of starvation induced tolerance, it would be interesting to see how removing the ability to induce that (via the pspA knock out) affects antibiotic uptake/efflux directly.

Response: Fig 4 depicts the experiment performed on wild type. We also tested antibiotic uptake in Δ pspA. Compared with wild type, intracellular BOCILLIN accumulation was found to increase to some extent in Δ pspA (Fig. l). As Reviewer 2 asked us to quantify fluorescent cells by flow cytometry (Reviewer2, Q16), we also performed flow cytometry to quantify intracellular BOCILLIN in Δ pspA. Water was used as negative control to confirm the FSC/SSC values of bacteria cells and PI was set as the targeted cells. The fluorescence intensity which is higher than 10^3 (P2) is 10.45% in the WT group, whereas P2 is 18.52% in the Δ pspA group (Fig. 4A-4D, Fig. S4B-S4F, manuscript).

Fig. (1) Confocal observation of intracellular BOCILLIN accumulation in wild type and $\Delta pspA$. The amount of intracellular BOCILLIN was more in $\Delta pspA$ than in wild type and intracellular BOCILLIN increased after CCCP treatment (scale bar: $4\mu\text{m}$).

Q3: Are there homologues of *pspA* that could be involved in PMF maintenance in the species shown in Figure 5?

Response: psp family genes are found mainly in E. coli and Y. enterocolitica. PspA is also involved in PMF maintenance in S. Typhimurium (PMID: 15853886). Even though they have not been identified in the bacterial species listed in Fig. 5, PMF is generally required in all bacterial species; recent publications report that compounds which target PMF exhibit bactericidal effect on strains of various bacterial species (PMID: 23972939, PMID: 30123191, PMID: 23441632).

Minor

MQ1: Mention which bacterium/strain is being used in the first paragraph of results.

Response: We have described the strain we used in the revised first paragraph of results. It was E. coli BW25113 which is the same strain described in Fig. 1-4. The knockout strains are also derived from BW25113.

MQ2: Experimental setups need to be clearer throughout the text. For example:

- a. In the first experiment to determine gene expression changes during starvations (beginning at line112) is there any antibiotic present?
- b. If not, please clarify the difference in the setup of this experiment and the experiment on the *psp* knock outs (starting on line 129).

Response: We have described the experimental setups in detail in the revised manuscript.

a. Yes. We added ampicillin $100\mu\text{g/ml}$ after resuspending the cells with saline. RNA Seq was

performed after the cells have been resuspended in saline containing ampicillin and then incubated at 37°C for 24 hrs.

MQ3: Explain the function of the *psp* operon when introduced (line 122).

*Response: *psp* operon is known to sense a change in PMF, membrane-stored curvature elastic stress, presence of mis-localized secretins and other factors; its activation renders bacterial cells able to maintain PMF or prevent mis-localized secretin toxicity. The *pspABCDE* operon is controlled by the regulatory protein PspF. PspA forms a complex with PspF to inhibit its activity. On one hand, PspA detects the inner membrane signals directly and interacts with the membrane to release PspF. On the other hand, PspB and PspC form an integral inner membrane complex. When secretin is mis-located into the inner membrane, the arrangement of PspB and PspC domains shifts so that PspA-binding site becomes available. PspA is sequestered by PspC, which releases active PspF to induce *pspABCDE* operon promoter (PMID: 16045608). Apart from the above, PspA binds to membrane phospholipids to prevent PMF dissipation (PMID: 17725563). We have explained the function and regulation of the *psp* operon / Psp response (MQ4) in the revised manuscript.*

MQ4: What is Psp response (line 146)?

*Response: The phage shock protein (Psp) response was identified as a response to phage infection in *E. coli*, but rather than being a specific response to phage, it is also elicited by conditions that undermines inner-membrane stability. The physiological role of Psp response mainly involves: (i) activities of the phage shock protein A which maintains the PMF, (ii) those of the phage shock proteins B and C to prevent mis-localized secretin toxicity. (PMID: 16045608).*

MQ5: Line 68: “high expression level of *tolC*”  “high expression of *tolC*”

Response: corrected accordingly.

MQ6: Line 93: lowercase a in *aeruginosa*

Response: corrected accordingly.

MQ7: Some figure captions list both the definition of two and four stars for significance when only one of these is needed.

Response: the error has been corrected.

MQ8: Fig. 2F: Why do some of the cells look like they have blunt ends and aren't nicely rod shaped?

Response: When preparing confocal samples, we put 2 μ L of stained cells (resuspended in saline

or PBS) onto slide and press the slide to make sure the cells were in thin layer so that we can obtain better vision. We observed that cells would display several overlapping layers if we did not press the slide well. This would make the cells dry quickly. To ensure consistency in experiments, we usually prepared four samples in one go (WT, WT+V, Δ pspA, Δ pspA+V), but then the observation time is lengthened. In that case the cells became dry and their morphology was distorted. We compared the bright field image of 100mM KCl-treated cells and non-KCl treated cells and found the morphology of 100mM KCl-treated cells were identical to that of the non-treatment group (Fig. j, Reviewer3, Q4). We counted the CFU of wild type without treatment for confocal microscopy and compared to those of cells after treatment. The CFU recorded in both cases were similar (Fig. i, Reviewer 3, Q2).

MQ9: There is a missing symbol showing up as a square in line 281 and multiple places throughout the supplementary materials.

Response: the error has been corrected.

MQ10: Line 441: Tolerant cells need not be dormant.

Response: corrected accordingly.

MQ11: Line 470: extra space before comma

Response: corrected accordingly.

MQ12: Supplementary line 33: rmp  rpm

Response: corrected accordingly.

Reviewers' comments:

Reviewer #1 (Remarks to the Author):

Summary

The revisions have greatly improved the paper. Below are comments that I believe will further enhance clarity and ease of understanding by readers. After these comments are addressed, I support publication of the article.

Comments

Line 73-76: citation required for point (ii).

Line 91-93 and Discussion: is the working model that modulation of PMF is a balancing act between initially adopting the tolerance phenotype (slight depletion of PMF), but not going so far as to abolish membrane polarization and preventing regrowth (excessive depletion of PMF)? If this is the model, I suggest highlighting this perhaps with Figure 6.

Line 145-146: emphasize (with literature citations) the importance of *pspA* relative to other genes in the *psp* gene family to ensure that readers have a better grasp of why you saw enhanced sensitivity in the *pspA* deletion mutant but none of the other *psp* genes.

Line 157-159: the *pspA* deletion sensitivity phenotype is quite clearly ampicillin specific – there is very little difference between the *pspA* deletion strain and wildtype upon exposure to gentamicin or ciprofloxacin. I think that this should be made even more clear in the paper – maybe even in the title, which could be changed to “Active Maintenance of Proton Motive Force is a Key Mechanism of Starvation-induced Bacterial Tolerance to Beta-lactams”. The fact that the *pspA* observations are ampicillin-specific are intriguing and suggests that other classes of bactericidal antibiotics will have their own unique genetic determinants of tolerance.

Line 207: the dye could not enter because the delta psi component of the PMF was depleted. DiSC3(5) required the delta psi component of the PMF to accumulate in the membrane. This needs to be made clear.

Figure 3b: why is CCCP a more potent potentiator than NaN₃? Can the authors provide rationale more clearly in the accompanying text of the manuscript?

Figure S3: the differing results with gentamicin and ciprofloxacin show that this is a case-by-case basis for *pspA* interactions, and no generalizations should be made. This is a comment that stems directly from an earlier suggestion.

Figure 4A and 4C: comparing these two plots, we only see a ~2-fold difference in BOCILLIN accumulation. Can this explain the multi-log difference in beta-lactam killing you observe earlier in the paper? Maybe worth a short mention.

Figure 4B and 4D: would you not expect to have the *pspA* deletion strain more sensitive to CCCP relative to wildtype, rather than wildtype having more BOCILLIN within the cells? I realize the difference is not large, but observable nonetheless and perhaps worth mentioning.

Figure 4E: it would be potentially useful to include the *pspA* deletion strain in this figure (+ and – CCCP, of course).

Line 456: it's hard to state confidently that this paper provides a comprehensive view of the role of PMF in tolerance. I suggest that the authors avoid overstating and remove this phrase.

Discussion: ensure that the discussion is focused on ampicillin and avoids over-generalizing to other antibiotics since the data with gentamicin and ciprofloxacin do not clearly align with those using ampicillin.

I believe that the images for Figure 6b and 6c are switched relative to the Figure legend.

Reviewer #2 (Remarks to the Author):

The authors considered by previous suggestions. I only have a few minor comments:

Fig. S3C: Given that the uptake of aminoglycosides is dependent on proton motive force, do your data in Fig. S3C suggest that CCCP increases the permeability of Gentamicin in the *pspA* mutant?

Line 428: please add a hyphen between "268" and "fold".

Line 453: please replace "Orman M. et al." with "Orman et al."

Reviewer #3 (Remarks to the Author):

The authors addressed the most of my concerns with revision of text and new experiments. The authors have also clarified their experimental procedures. I believe that the manuscript has substantially improved its quality. However, I am still concerned about the method taken for microscopy experiments. The authors described in their response that the strange morphological appearance of cells in their microscopy data is due to drying of sample. Seeing such a significant morphological impact, it seems reasonable to assume other cellular properties are also affected in these samples, which undermines the confidence in their conclusions derived by microscopy data. I would like to recommend the authors to repeat the microscopy experiments with cells placed on agarose (or agar) block. A more detailed description of this method can be found in <https://pubmed.ncbi.nlm.nih.gov/22179594/>. This method should prevent samples from drying.

Reviewer #4 (Remarks to the Author):

The authors have addressed the points I raised in my original review.

Responses to reviewers' comments

Reviewer #1

Q1: Line 73-76: citation required for point (ii).

Response: A reference has been added.

Q2: Line 91-93 and Discussion: is the working model that modulation of PMF is a balancing act between initially adopting the tolerance phenotype (slight depletion of PMF), but not going so far as to abolish membrane polarization and preventing regrowth (excessive depletion of PMF)? If this is the model, I suggest highlighting this perhaps with Figure 6.

Response: Yes. Cells would die if PMF is completely abolished by treatment with compounds mentioned in Line 92. Tolerant cells maintained a basic PMF level and would die upon exposure to compounds which cause further dissipation of PMF. We have highlighted this in Figure 6.

Q3: Line 145-146: emphasize (with literature citations) the importance of *pspA* relative to other genes in the *psp* gene family to ensure that readers have a better grasp of why you saw enhanced sensitivity in the *pspA* deletion mutant but none of the other *psp* genes.

Response: Modified as suggested.

Q4: Line 157-159: the *pspA* deletion sensitivity phenotype is quite clearly ampicillin specific – there is very little difference between the *pspA* deletion strain and wildtype upon exposure to gentamicin or ciprofloxacin. I think that this should be made even more clear in the paper – maybe even in the title, which could be changed to “Active Maintenance of Proton Motive Force is a Key Mechanism of Starvation-induced Bacterial Tolerance to Beta-lactams”. The fact that the *pspA* observations are ampicillin-specific are intriguing and suggests that other classes of bactericidal antibiotics will have their own unique genetic determinants of tolerance.

*Response: The title has been changed to reflect the role of PspA-dependent mechanisms in specifically maintaining tolerance to β -lactam antibiotics. The survival population in Δ *pspA* after exposure to gentamicin is lower than in wild type (Fig. S1C), even though the difference between them is not great. The survival population after CCCP and gentamicin-combined treatment also decreased when compared with gentamicin treatment alone (Fig. S3B&C). So we modified the description as “we found that the size of the tolerant Δ *pspA* population during*

gentamicin treatment was only slightly less than that of the wild type strain. The tolerant Δ pspA population against ciprofloxacin was similar to that of the wild type strain, suggesting that the pspA gene product specifically enhances tolerance to β -lactams” in the revised manuscript.

Q5: Line 207: the dye could not enter because the delta psi component of the PMF was depleted. DiSC3(5) required the delta psi component of the PMF to accumulate in the membrane. This needs to be made clear.

Response: delta psi is the term used to describe transmembrane electric potential. We also called it membrane potential for short (PMID: 27148531, PMID: 20566861). We described in the manuscript the reason why DiSC3(5) could not enter Δ pspA is that the membrane potential was too low.

Q6: Figure 3b: why is CCCP a more potent potentiator than NaN₃? Can the authors provide rationale more clearly in the accompanying text of the manuscript?

Response: CCCP is a protonophore that increases membrane permeability to protons, leading to a disruption of membrane potential (PMID: 29277693). NaN₃ inhibited cytochrome c oxidase, the final enzyme in electron transport chain to prevent transmembrane potential (PMID: 8380332). PspA forms oligomers in the inner membrane to prevent proton leakage (PMID: 27297125). The effect of CCCP is so strong that it can dissipate PMF even when PMF generation mechanism and PspA function are effective, whereas NaN₃ targets on the last step of electron transport chain and does not affect the function of PspA or cause direct dissipation of PMF. In addition, the membrane potential can still be generated by the first three steps of electron transport chain even in the presence of NaN₃. Hence the effect of PMF-dissipating effect of CCCP is much stronger than NaN₃. The respective functions of NaN₃ and CCCP have been clearly described in the revised manuscript.

Q7: Figure S3: the differing results with gentamicin and ciprofloxacin show that this is a case-by-case basis for pspA interactions, and no generalizations should be made. This is a comment that stems directly from an earlier suggestion.

Response: Corrected accordingly.

Q8: Figure 4A and 4C: comparing these two plots, we only see a ~2-fold difference in BOCILLIN accumulation. Can this explain the multi-log difference in beta-lactam killing you observe earlier in the paper? Maybe worth a short mention.

Response: The difference in BOCILLIN accumulation is consistent with the amount of tolerant cells that survived different treatments (CCCP in combination with ampicillin in Fig. 3 A&B, ampicillin treatment in Δ pspA in Fig.1A). ~2 fold difference in

BOCILLIN accumulation was observed between wild type and Δ pspA, whereas significant difference in *BOCILLIN* accumulation was observed between no-CCCP treatment and CCCP treatment (Fig. 4). The survival population decreased significantly after treatment with the CCCP and ampicillin combination (Fig. 3 A&B), The size of the tolerant population that survived treatment with ampicillin in Δ pspA was much higher than that recorded in the combined CCCP and ampicillin treatment (Fig. 1A). Hence the relative decrease in size of the tolerant population recorded in these assays is consistent with the *BOCILLIN* accumulation data. We have described and discussed these findings in the revised manuscript.

Q9: Figure 4B and 4D: would you not expect to have the pspA deletion strain more sensitive to CCCP relative to wildtype, rather than wildtype having more *BOCILLIN* within the cells? I realize the difference is not large, but observable nonetheless and perhaps worth mentioning.

Response: CCCP increases membrane permeability to proton and PspA forms oligomers in the inner membrane to prevent proton leakage (Q6). In both wild type and Δ pspA, CCCP can effectively and significantly increase membrane permeability to proton so we can observed a large amount of BOCILLIN accumulation in both of these two strains but cannot observe much difference in BOCILLIN accumulation between these two strains (97.86% in wild type and 91.25% in Δ pspA). Apart from comparing the percentage of cells whose BOCILLIN intensity $>10^3$ RFU, the amount of cells that exhibiting extremely high BOCILLIN intensity ($>2 \times 10^4$ RFU) also deserves attention. The percentage of cells exhibiting extremely high BOCILLIN intensity in CCCP treated Δ pspA group was 25.34%, while that in CCCP-treated wild type was 9.76%. We have mentioned this finding in the reversed manuscript.

Q10: Figure 4E: it would be potentially useful to include the pspA deletion strain in this figure (+ and – CCCP, of course).

Response: We have added the pspA deletion strain in the revised manuscript. Nile Red fluorescence intensity of Δ pspA was higher than wild type, indicating an attenuated efflux efficiency in Δ pspA. This is consistent with our conclusion that PMF dissipation negatively affects efflux activity.

Q11: Line 456: it's hard to state confidently that this paper provides a comprehensive

view of the role of PMF in tolerance. I suggest that the authors avoid overstating and remove this phrase.

Response: Modified accordingly.

Q12: Discussion: ensure that the discussion is focused on ampicillin and avoids over-generalizing to other antibiotics since the data with gentamicin and ciprofloxacin do not clearly align with those using ampicillin.

Response: Modified accordingly.

Q13: I believe that the images for Figure 6b and 6c are switched relative to the Figure legend.

Response: Corrected accordingly.

Reviewer #2

Q1: Fig. S3C: Given that the uptake of aminoglycosides is dependent on proton motive force, do your data in Fig. S3C suggest that CCCP increases the permeability of Gentamicin in the *pspA* mutant?

Response: Our data showed that CCCP potentiated gentamicin killing effect upon tolerant cells. Apart from CCCP, HT61 which dissipates membrane potential is reported to enhance killing effect of gentamicin upon MRSA (PMID: 23042813). PAAG which dissipates membrane potential is also reported to kill tolerant cells together with tobramycin (Fig. 2A, PMID: 30123191). Electric current and electrochemical scaffold which enhance membrane permeability are reported to potentiate the killing effect of tobramycin (PMID: 28649396, PMID: 22840233, PMID: 26461119). Our data are consistent with these findings and other viewpoint cannot be confirmed according to the existing data.

Q2: Line 428: please add a hyphen between "268" and "fold".

Response: Modified accordingly.

Q3: Line 453: please replace "Orman M. et al." with "Orman et al."

Response: Modified accordingly.

Reviewer #3

Q1: The authors addressed the most of my concerns with revision of text and new

experiments. The authors have also clarified their experimental procedures. I believe that the manuscript has substantially improved its quality. However, I am still concerned about the method taken for microscopy experiments. The authors described in their response that the strange morphological appearance of cells in their microscopy data is due to drying of sample. Seeing such a significant morphological impact, it seems reasonable to assume other cellular properties are also affected in these samples, which undermines the confidence in their conclusions derived by microscopy data. I would like to recommend the authors to repeat the microscopy experiments with cells placed on agarose (or agar) block. A more detailed description of this method can be found in <https://pubmed.ncbi.nlm.nih.gov/22179594/>. This method should prevent samples from drying.

Response: The microscopy experiment has been repeated with cells embedded in agarose block. The strange morphology due to the drying effect disappeared. Thanks for the advice, the result has been included in the revised manuscript.

REVIEWERS' COMMENTS:

Reviewer #1 (Remarks to the Author):

The manuscript is improved.

One point: in Figure 1A, what we are most interested in is the ratio of viable cells +/- ampicillin. In other words the viability ratio (WT+AMP / WT) relative to (dpspA+AMP / dpspA) at endpoint time. This would highlight more clearly the decreased tolerance by deleting pspA. Perhaps this similar analysis could run through all relevant panels in the paper, although a focus on Figure 1A is most useful.

Reviewer #3 (Remarks to the Author):

The microscopy image still look strange to me, but other data are strongly supporting the conclusion of the paper. Hence, I recommend this manuscript to be accepted.